# Zero-shot Denoising via Neural Compression: Theoretical and algorithmic framework

**Ali Zafari**[*]
ali.zafari@rutgers.edu

**Xi Chen**[*]
xi.chen15@rutgers.edu

**Shirin Jalali**
shirin.jalali@rutgers.edu

Rutgers University
New Brunswick, NJ, USA

## Abstract

Zero-shot denoising aims to denoise observations without access to training samples or clean reference images. This setting is particularly relevant in practical imaging scenarios involving specialized domains such as medical imaging or biology. In this work, we propose the *Zero-Shot Neural Compression Denoiser* (ZS-NCD), a novel denoising framework based on neural compression. ZS-NCD treats a neural compression network as an untrained model, optimized directly on patches extracted from a single noisy image. The final reconstruction is then obtained by aggregating the outputs of the trained model over overlapping patches. Thanks to the built-in entropy constraints of compression architectures, our method naturally avoids overfitting and does not require manual regularization or early stopping. Through extensive experiments, we show that ZS-NCD achieves state-of-the-art performance among zero-shot denoisers for both Gaussian and Poisson noise, and generalizes well to both natural and non-natural images. Additionally, we provide new finite-sample theoretical results that characterize upper bounds on the achievable reconstruction error of general maximum-likelihood compression-based denoisers. These results further establish the theoretical foundations of compression-based denoising. Our code is available at: https://github.com/Computational-Imaging-RU/ZS-NCDenoiser.

## 1 Introduction

**Background and motivation** Denoising is a fundamental problem in classical signal processing and has recently gained renewed attention from the machine learning community. Let $\boldsymbol{x} = (x_1, \ldots, x_n) \in \mathbb{R}_+^n$ denote a non-negative signal of length $n$, where signal $\boldsymbol{x}$ is not observable in many systems. Instead, we observe a noisy version $\boldsymbol{y} = (y_1, \ldots, y_n)$, where the observations are conditionally independent given $\boldsymbol{x}$, and each entry is distributed according to a common conditional distribution:

$$\boldsymbol{y} \sim \prod_{i=1}^{n} p(y_i \mid x_i).$$

We assume that the noise mechanism is memoryless (independent across coordinates) and homogeneous (identical across entries). The goal of a denoising algorithm is to estimate $\boldsymbol{x}$ from the noisy observations $\boldsymbol{y}$. Given its prevalence in imaging and data acquisition systems, denoising has been a central topic in signal processing for decades. Classical denoising methods rely on explicit structural assumptions about the underlying signal $\boldsymbol{x}$, often hand-crafted by domain experts [1, 2, 3, 4, 5, 6, 7, 8]. In contrast, recent advances in machine learning have enabled a new class of data-driven denoising

---

[*]Equal contribution.

39th Conference on Neural Information Processing Systems (NeurIPS 2025).

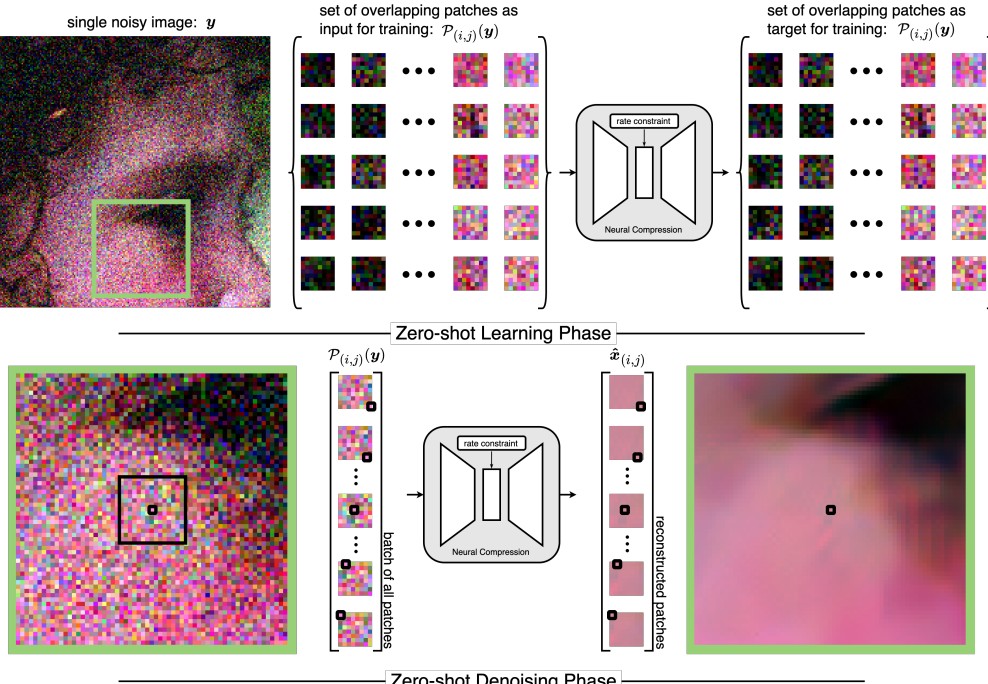

Figure 1: Zero-Shot Neural Compression Denoiser (ZS-NCD). Learning phase: a neural compression model (architecture shown in Fig. 5 of the supplementary material) is trained on overlapping patches extracted from a single noisy image. Denoising phase: each pixel is reconstructed by averaging predictions across neighboring patches processed by the trained model.

algorithms. These methods learn the optimal denoising function from data, leveraging statistical patterns directly from signal and noise distributions.

While learning-based approaches achieve state-of-the-art performance and often outperform classical methods in controlled settings, they face significant challenges in practice:

1. Supervision requirement: Most learning-based methods require training set of paired samples $\{(\boldsymbol{y}_i, \boldsymbol{x}_i)\}_{i=1}^{m}$, where $\boldsymbol{x}_i$ is clean signal and $\boldsymbol{y}_i$ is its noisy counterpart. In practical scenarios such as medical imaging, $\{\boldsymbol{x}_i\}_{i=1}^{m}$ are unavailable or prohibitively expensive to obtain.

2. Data efficiency: These methods usually need lots of training data. Acquiring sufficient samples is difficult or costly, particularly in domains with strict data acquisition constraints.

To mitigate the reliance on paired clean and noisy samples, several self-supervised denoising methods have been developed that learn directly from noisy observations, without access to clean ground truth signals [9, 10, 11, 12, 13]. While these approaches alleviate the supervision requirement, they typically depend on access to large collections of noisy data and often yield suboptimal performance compared to methods trained with clean targets. Moreover, the absence of clean supervision necessitates the use of complex neural architectures and training schemes, which can make these methods computationally demanding and difficult to optimize in practice.

These challenges have sparked growing interest in *zero-shot denoisers*, which aim to recover clean signals from noisy observations without access to paired data or extensive noisy training data. Such methods are particularly appealing in domains where acquiring clean data is infeasible, and they offer the potential for deployable denoisers that adapt to individual inputs with general purpose.

**From neural compression to zero-shot denoising** Denoising algorithms—ranging from classical signal processing techniques to deep learning methods—fundamentally rely on the assumption that real-world signals are highly structured. Compression-based denoising leverages this same principle, but rather than directly solving the inverse problem, it instead performs lossy compression on the

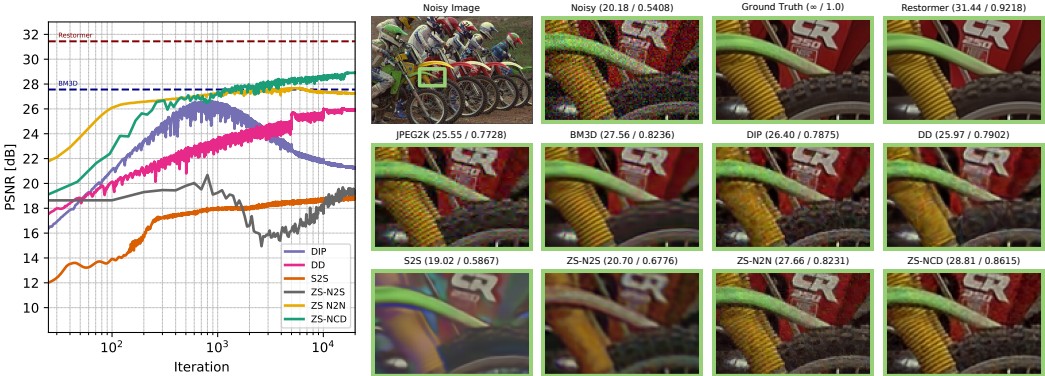

Figure 2: Zero-shot denoising of *Kodim05* with AWGN ($\sigma = 25$). **Left**: PSNR versus training iterations for zero-shot denoisers. Performance of BM3D [18] and Restormer [19] are included as a classical baseline and as a supervised empirical upper bound, respectively. **Right**: Visual reconstructions with PSNR above each image. Compression-based denoising based on JPEG-2K [20] achieves inferior performance. Learning-based zero-shot denoisers often struggle with either overfitting or high bias. DIP [21] and DD [22] require early stopping to avoid overfitting. ZS-N2S [12] and S2S [23] struggle with high-resolution color images, and ZS-N2N [24] often produces noisy outputs with potential overfitting. BM3D tends to oversmooth the denoised image. In contrast, ZS-NCD avoids these issues.

noisy observation $y$, under the hypothesis that the clean signal $x$ lies in a lower-complexity subspace and is therefore more compressible.

In lossy compression, the goal is to represent signals from a target class using discrete encodings with minimal distortion. When applied to noisy data, the intuition is that a lossy compressor—operating at a distortion level matched to the noise—will favor reconstructions close to the original clean signal. While this approach has a strong theoretical foundation [14, 15], classical compression-based denoisers have shown limited empirical success, particularly for natural image denoising.

In this work, we revisit this idea in light of recent progress in *neural compression*, where learned encoders and decoders have demonstrated strong rate-distortion performance across a variety of image domains [16, 17]. Building on this foundation, we propose a *zero-shot* denoising method that we call the **Zero-Shot Neural Compression Denoiser (ZS-NCD)**. Unlike traditional neural compression models that are trained on large corpora of clean high-resolution images, ZS-NCD learns directly from a single noisy input image. Specifically, we extract overlapping patches from the noisy image, and train a neural compression network on those patches alone—without any clean supervision or prior dataset. Once trained, the denoiser is applied to all patches from the same image, and the final output is obtained by averaging the predictions in overlapping regions. This approach is illustrated in Figure 1.

Despite relying solely on the noisy input and operating without supervision, ZS-NCD achieves state-of-the-art performance among zero-shot denoising methods across diverse noise models, and remains robust even on inputs that lie outside the natural image distribution. We compare it against the baselines in Figure 2, ZS-NCD shows superior performance in denoising and training stability.

**Paper contributions**    This paper introduces a zero-shot image denoising framework based on neural compression. Our main contributions are:

- **A zero-shot denoising algorithm using neural compression.** We propose a fully unsupervised method that trains a neural compression network on image patches of the noisy input. It does not rely on clean images, paired datasets, or prior training on the target distribution. It is architecture-agnostic and leverages only the structure present in the observed noisy image.

- **Theoretical results connecting denoising and compression performance.** We establish finite-sample upper bounds on the reconstruction error of the proposed compression-based maximum likelihood denoisers, for both Gaussian and Poisson noise models.

- **Extensive empirical validation.** We demonstrate that our method achieves state-of-the-art performance among zero-shot denoising techniques across a range of noise models and datasets.

## 2  Related work

**Self-supervised and zero-shot denoising**  Supervised learning-based denoisers such as DnCNN [25] and Restormer [19] achieve state-of-the-art performance across various noise models, but require large datasets of paired clean and noisy images—often impractical in real-world settings. To avoid clean images, self-supervised methods have been proposed, including Noise2Noise [10], Noise2Self [12], Noise2Void [11], Noise2Same[26] and Noise2Score [13], which only use noisy images for training. However, their reliance on large noisy datasets remains a limitation. Zero-shot denoisers address this by training on a single noisy image. These include (i) untrained networks like DIP [27] and Deep Decoder [22], and (ii) single-image adaptations of self-supervised methods, *e.g.,* N2F [28], ZS-N2N [24], ZS-N2S [12] and its augmented variant with ensembling, S2S [23]. More recently, DS-N2N [29] improves ZS-N2N by further upsampling the downsampled paired noisy images. Pixel2Pixel [30] boosts the performance by using non-local similarity approach. DIP-based models avoid masking and leverage full-image context, but require early stopping or under-parameterization to avoid overfitting. Self-supervised variants suffer from masking-induced information loss. Hybrid approaches, such as masked pretraining-based method [31], uses external datasets for training and perform zero-shot inference, thus falling outside the zero-shot setting studied here.

**Neural compression**  Learning-based lossy compression, often referred to as neural compression, uses an autoencoder architecture combined with an entropy model to estimate and constrain the bitrate at the bottleneck [32, 33]. These methods have significantly outperformed traditional codecs, particularly in image [32, 34, 35, 36, 37] and video [38, 39, 40] compression. In addition to these standard settings, several works have also explored applying neural compression to noisy data, either by adapting neural compression models for more efficient encoding of noisy images [41, 42, 43, 44].

**Compression-based denoising**  Compression-based denoising leverages the insight that structured signals are inherently more compressible than their noisy counterparts. This connection was formalized by Donoho [14], who introduced the minimum Kolmogorov complexity estimator, and further refined by Weissman et al. [15], showing that, under certain conditions on both the signal and the noise, *optimal* lossy compression of a noisy signal-followed by suitable post-processing-can asymptotically achieve optimal denoising performance. Prior to these theoretical developments, early empirical methods such as wavelet-based schemes [45, 46, 47] and MDL-inspired heuristics [48] had explored this principle. Nevertheless, traditional compression-based denoisers have generally underperformed in high-dimensional settings such as image denoising.

Learning-based joint compression and denoising using neural compression has been explored in recent works [49, 50], where the goal is to achieve lower rate in compression. The empirical application of training neural compression for AWGN denoising was also proposed in [51]. Training the neural compression models in these works requires a dataset of images. In contrast, our proposed ZS-NCD is a two-step denoiser based on neural compression, trained on a single noisy image. It achieves state-of-the-art performance across both AWGN and Poisson noise models. Moreover, we contribute new theoretical results that advance the foundations of compression-based denoising.

## 3  Compression-based denoising: Theoretical foundations

**Lossy compression**  Let $\mathcal{Q} \subset \mathbb{R}^n$ denote the signal class of interest, such as vectorized natural images of a fixed size. A lossy compression code for $\mathcal{Q}$ is defined by an encoder-decoder pair $(f, g)$, $f : \mathcal{Q} \to \{1, \ldots, 2^R\}$, and $g : \{1, \ldots, 2^R\} \to \mathbb{R}^n$. The performance of a lossy code is characterized by: i) *Rate $R$*, indicating the number of distinct codewords; ii) *Distortion $\delta$*, defined as the worst-case per-symbol mean squared error (MSE) over the signal class:

$$\delta = \sup_{\boldsymbol{x} \in \mathcal{Q}} \frac{1}{n} \|\boldsymbol{x} - g(f(\boldsymbol{x}))\|_2^2.$$

The set of reconstructions produced by the decoder forms the codebook:

$$\mathcal{C} = \{g(i) : i = 1, \ldots, 2^R\} \subset \mathbb{R}^n.$$

**Compression-based denoising**    We propose compression-based denoising as a structured maximum likelihood (ML) estimation. Given a noisy observation $\boldsymbol{y} \sim \prod_{i=1}^{n} p(y_i \mid x_i)$ and a a lossy compression code $(f, g)$ for $\mathcal{Q}$, the compression-based ML denoiser solves

$$\hat{\boldsymbol{x}} = \arg\min_{\mathbf{c} \in \mathcal{C}} \mathcal{L}(\mathbf{c}; \boldsymbol{y}), \quad \text{where} \quad \mathcal{L}(\mathbf{c}; \boldsymbol{y}) := -\sum_{i=1}^{n} \log p(y_i \mid c_i).$$

This formulation leverages the fact that clean signals, by virtue of their structure, are more compressible than their noisy counterparts. Therefore, the most likely codeword under the noise model, when selected from a codebook designed to represent clean signals, serves as a natural denoising estimate. This ML-based view unifies denoising across noise models and provides a principled way to select reconstructions from a discrete, structure-aware prior.

In the case of AWGN: $\boldsymbol{y} = \boldsymbol{x} + \boldsymbol{z}$, where $\boldsymbol{z} \sim \mathcal{N}(\mathbf{0}, \sigma_z^2 I_n)$, the described denoiser simplifies to:

$$\hat{\boldsymbol{x}} = \arg\min_{\mathbf{c} \in \mathcal{C}} \|\boldsymbol{y} - \mathbf{c}\|_2^2. \tag{1}$$

That is, denoising corresponds to projecting the noisy observation onto the nearest codeword.

Poisson noise commonly arises in low-light and photon-limited imaging scenarios. In this setting, each $y_i$ is modeled as a Poisson random variable with mean $\alpha x_i$: $y_i \sim \text{Poisson}(\alpha x_i)$. Under this model, the compression-based ML denoiser simplifies to

$$\hat{\boldsymbol{x}} = \arg\min_{\mathbf{c} \in \mathcal{C}} \sum_{i=1}^{n} \left( \alpha c_i - y_i \log c_i \right). \tag{2}$$

While the loss function in (2) is statistically well-motivated, it is more sensitive to optimization issues than its Gaussian counterpart due to the curvature and nonlinearity of the log term. To improve robustness and simplify optimization, we also consider an alternative loss based on a normalized squared error between **c** and the rescaled observations:

$$\hat{\boldsymbol{x}} = \arg\min_{\mathbf{c} \in \mathcal{C}} \left\| \mathbf{c} - \tfrac{1}{\alpha} \boldsymbol{y} \right\|_2^2. \tag{3}$$

**Theoretical analysis**    We begin by analyzing the performance of compression-based ML denoising under AWGN. The following result provides a non-asymptotic upper bound on the reconstruction error in terms of the compression rate and distortion. All proofs can be found in Appendix A.

**Theorem 1.** *Assume that $\boldsymbol{x} \in \mathcal{Q}$ and let $(f, g)$ denote a lossy compression for $\mathcal{Q}$ that operates at rate $R$ and distortion $\delta$. Consider $\boldsymbol{y} = \boldsymbol{x} + \boldsymbol{z}$, where $\boldsymbol{z} \sim \mathcal{N}(\mathbf{0}, \sigma_z^2 I_n)$. Let $\hat{\boldsymbol{x}}$ denote the output of the compression-based denoiser defined by $(f, g)$ as in (1). Then,*

$$\frac{1}{\sqrt{n}} \|\boldsymbol{x} - \hat{\boldsymbol{x}}\|_2 \leq \sqrt{\delta} + 2\sigma_z \sqrt{\frac{(2 \ln 2) R}{n}} (1 + 2\sqrt{\eta}), \tag{4}$$

*with a probability larger than $1 - 2^{-\eta R + 2}$.*

This bound decomposes the denoising error into two terms: a *distortion term* $\sqrt{\delta}$, which reflects the approximation quality of the compression code, and a *rate-dependent term* that scales with the square root of the code rate $R$. The latter captures the likelihood concentration around the clean signal in high-probability regions of the noise distribution. Notably, the result holds non-asymptotically and does not assume the code is optimal, only that it provides a distortion-$\delta$ covering of $\mathcal{Q}$. This highlights that even non-ideal compression codes can enable effective denoising, provided the rate-distortion tradeoff is well-calibrated.

To better understand the implications of Theorem 1, in the following corollary, we focus on the special case of $k$-sparse signals.

**Corollary 1** (AWGN, sparse signals)**.** *Let $\mathcal{Q}_n$ denote the set of $k$-sparse vectors in $\mathbb{R}^n$ satisfying $\|\boldsymbol{x}\|_\infty \leq 1$. Fix a parameter $\eta \in (0, 1)$, and suppose $\boldsymbol{y} = \boldsymbol{x} + \boldsymbol{z}$ where $\boldsymbol{z} \sim \mathcal{N}(0, \sigma_z^2 I_n)$. Then, there exists a family of compression codes such that, when used with the denoiser defined in (1), with a probability larger than $1 - \mathrm{e}^{-\eta k \log(n/k)}$, the estimate $\hat{\boldsymbol{x}}$ satisfies*

$$\frac{1}{\sqrt{n}} \|\boldsymbol{x} - \hat{\boldsymbol{x}}\|_2 \leq \sigma_z C \sqrt{\frac{k}{n} \log_2(\frac{n}{k}) + \gamma_n} + \frac{1}{\sqrt{n}}, \tag{5}$$

*where $C = 2(1 + 2\sqrt{\eta})\sqrt{2 \ln 2}$ and $\gamma_n = \frac{k \log_2 k}{2n} + \frac{\log_2 k + k(\log_2 \mathrm{e} + 1)}{n}$.*

Corollary 1 provides a high-probability bound on the normalized error $\frac{1}{\sqrt{n}}\|\hat{\boldsymbol{x}} - \boldsymbol{x}\|_2$. Squaring both sides and having $(\sqrt{a} + \sqrt{b})^2 \leq 2(a + b)$, it follows that, with high probability

$$\frac{1}{n}\|\hat{\boldsymbol{x}} - \boldsymbol{x}\|_2^2 \leq 2\sigma_z^2 C^2 \Big(\frac{k}{n}\log_2(\frac{n}{k}) + \gamma_n\Big) + \frac{2}{n}.$$

Thus, up to universal constants, the dominant term in the upper bound scales as $\sigma_z^2 \frac{k}{n}\log(\frac{n}{k})$. This matches the known minimax rate for estimating $k$-sparse signals in Gaussian noise when $k/n \to 0$; see [52]. Determining whether the residual term $\frac{2}{n}$ is an artifact of our proof or a real barrier to optimality is an interesting problem. Finally, while the comparison above is high-probability rather than in expectation, one can integrate the tail bound from the proof to obtain an expected-risk bound.

We next extend our analysis to signal-dependent noise model. Poisson noise is particularly relevant in imaging applications such as microscopy and astronomy, where photon counts vary with signal intensity. Unlike Gaussian noise, Poisson observations induce a non-linear likelihood surface, making analysis more delicate. Theorem 2 and 3 establish performance guarantees for compression-based Poisson denoising, using both exact ML formulation and a practical squared-error surrogate.

**Theorem 2.** *Consider the same setup of lossy compression as in Theorem 1. Assume that for any $\boldsymbol{x} \in \mathcal{Q}$, $x_i \in (x_{\min}, x_{\max})$, where $0 < x_{\min} < x_{\max} < 1$. Assume that $y_1, \ldots, y_n$ are independent with $y_i \sim \text{Poisson}(\alpha x_i)$. Let $\hat{\boldsymbol{x}}$ denote the solution of (2). Let $C_1 = x_{\max}^5/(x_{\min}^2)$ and $C_2 = \frac{x_{\max}^2}{x_{\min}^3}\beta\sqrt{(\frac{4}{\ln 2})}(\sqrt{1 + \eta} + \sqrt{\eta})$. Then, with a probability larger than $1 - 2^{-\eta R + 2}$,*

$$\frac{1}{n}\|\boldsymbol{x} - \hat{\boldsymbol{x}}\|_2^2 \leq C_1\delta + C_2\sqrt{\frac{R}{n\alpha}}. \tag{6}$$

**Theorem 3.** *Consider the same setup as in Theorem 2. Let $\hat{\boldsymbol{x}}$ denote the solution of (3). Let $C = 4\sqrt{\ln 2}(\sqrt{1 + \eta} + \sqrt{\eta} + 1)$. Then, with a probability larger than $1 - 2^{-\eta R + 2}$,*

$$\frac{1}{n}\|\boldsymbol{x} - \hat{\boldsymbol{x}}\|_2^2 \leq \delta + C\sqrt{\frac{R}{n\alpha}}. \tag{7}$$

**Remark 1.** *Theorems 2 and 3 show that, in the case of Poisson noise, minimizing either the ML loss function or the computationally efficient MSE loss function can recover the signal. This result is also consistent with our simulations reported later in Section 5.*

## 4 Zero-shot compression-based denoiser

We refer to a general class of learning-based denoisers that operate by compressing noisy images using neural compression as the Neural Compression Denoiser (NCD). In this framework, denoising is achieved by identifying a low-complexity reconstruction from the output of a neural compression model. In the previous section, we characterized the performance of such denoisers in a setting where the compression code is fixed in advance, either learned from external data or designed using classical methods, and applied independently of the noisy input. This setup is not zero-shot, as it relies on prior training or code design. Inspired by this idea, we now propose a fully unsupervised variant: the **Zero-Shot Neural Compression Denoiser (ZS-NCD)**. In ZS-NCD, a neural compression network is trained directly on patches extracted from a single noisy image, without access to clean targets or external data. This section describes the ZS-NCD architecture and optimization procedure in detail.

**Proposed zero-shot denoiser: ZS-NCD** Let $\mathcal{P}_{(i,j)} : \mathbb{R}^{h \times w} \to \mathbb{R}^{k \times k}$ denote the patch extraction operator, which returns a $k \times k$ patch whose top-left corner is at pixel $(i, j)$. Let $f_{\boldsymbol{\theta}_1}$ and $g_{\boldsymbol{\theta}_2}$ denote the encoder and decoder networks, parameterized by weights $\boldsymbol{\theta}_1$ and $\boldsymbol{\theta}_2$, respectively. Define $\mathcal{I}$ as the set of all coordinates $(i, j) \in \{1, \ldots, h - k + 1\} \times \{1, \ldots, w - k + 1\}$ from which a valid $k \times k$ patch can be extracted.

Given a single noisy image $\boldsymbol{y}$, the ZS-NCD is trained to minimize the following patchwise objective:

$$(\hat{\boldsymbol{\theta}}_1, \hat{\boldsymbol{\theta}}_2) = \arg\min_{(\boldsymbol{\theta}_1, \boldsymbol{\theta}_2)} \sum_{(i,j) \in \mathcal{I}} \Big(\mathcal{L}_K(g_{\boldsymbol{\theta}_2}(f_{\boldsymbol{\theta}_1}(\mathcal{P}_{(i,j)}(\boldsymbol{y}))), \mathcal{P}_{(i,j)}(\boldsymbol{y})) - \lambda \log \mathbb{P}\big(f_{\boldsymbol{\theta}_1}(\mathcal{P}_{(i,j)}(\boldsymbol{y}))\big)\Big), \tag{8}$$

where $\mathbb{P}\big(f_{\boldsymbol{\theta}_1}(\mathcal{P}_{(i,j)}(\boldsymbol{y}))\big)$ denotes the likelihood (or entropy model) of the latent code produced by the encoder, and $\lambda > 0$ is a hyperparameter controlling the trade-off between fidelity and compressibility.

**Algorithm 1** Finding Lagrangian coefficient $\lambda$

1: **Initialize:** $\lambda^{(0)} > 0$, $\texttt{tol} > 0$, $\zeta \in (0,1)$
2: **for** $k = 0, 1, \ldots, K_{\max} - 1$ **do**
3:     Estimate $\hat{\boldsymbol{x}}^{(k)}$ from (9).
4:     Compute residual $r^{(k)} = \|\hat{\boldsymbol{x}}^{(k)} - \boldsymbol{y}\|^2$.
5:     Compute scale factor $\beta = (r^{(k)} - n\sigma_z^2)/n\sigma_z^2$.
6:     **if** $|\beta| \le \texttt{tol}$ **then**
7:       **break**
9:     **else**
8:
$$\lambda^{(k+1)} := \begin{cases} \lambda^{(k)}/(1 + \zeta|\beta|), & \beta > 0 \\ \lambda^{(k)}(1 + \zeta|\beta|), & \text{otherwise} \end{cases}$$
10:     **end if**
11: **end for**
12: **Return:** $\lambda^* = \lambda^{(k)}$

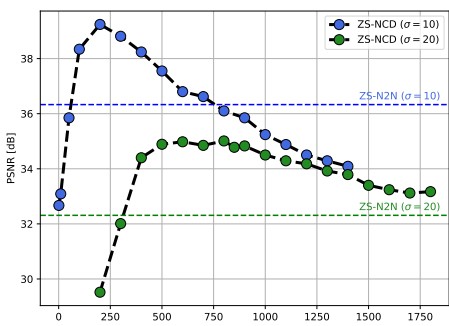

Figure 3: Effect of $\lambda$ in denoising Mouse Nuclei image.

In (8), $K = k^2$, and the function $\mathcal{L}_K : \mathbb{R}^K \times \mathbb{R}^K \to \mathbb{R}_+$ is a distortion loss determined by the noise model, as defined in Section 3. For example, in the AWGN case, $\mathcal{L}_K$ corresponds to the squared $\ell_2$ norm of the distance between a noisy patch and its neural compression reconstruction. Note that $f_{\boldsymbol{\theta}_1}$ maps the input into a discrete latent space, which is non-differentiable and thus incompatible with standard gradient-based optimization. To address this, we follow the neural compression framework of [32], using a continuous relaxation during training (e.g., uniform noise injection) and applying actual discretization only at test time. The entropy term $\mathbb{P}$ is modeled using a factorized, non-parametric density [34].

After training, the denoised image is obtained by applying the encoder and decoder to each patch and averaging the overlapping outputs. For each pixel $(i, j)$, let $\mathcal{I}_{(i,j)} \subset \mathcal{I}$ denote the set of patch locations such that $\mathcal{P}_{(i',j')}$ includes the pixel $(i, j)$. The final estimate at location $(i, j)$ is given by

$$\hat{x}_{(i,j)} = \frac{1}{|\mathcal{I}_{(i,j)}|} \sum_{(i',j') \in \mathcal{I}_{(i,j)}} g_{\boldsymbol{\theta}_2}(f_{\boldsymbol{\theta}_1}(\mathcal{P}_{(i',j')}(\boldsymbol{y})))\big|_{(i-i', j-j')}, \tag{9}$$

where $|\mathcal{I}_{(i,j)}|$ denotes the number of patches covering pixel $(i, j)$, and $\cdot|_{(a,b)}$ denotes the $(a, b)$-th pixel of the patch output. For interior pixels away from the boundary, $|\mathcal{I}_{(i,j)}| = K$. As shown later in Section 5, this aggregating of reconstructed patches significantly enhances denoising performance.

**Setting the hyperparameter $\lambda$** The ZS-NCD objective in (8) includes a hyperparameter, $\lambda$, which balances reconstruction fidelity and compressibility. Interpreted through the lens of lossy compression, varying $\lambda$ allows the model to explore different rate-distortion trade-offs. However, in the context of denoising, our goal is not compression but accurate signal recovery from the noisy observation $\boldsymbol{y}$. This raises the central question: *how should $\lambda$ be selected to optimize denoising performance?* In the following, we explain our approach for setting $\lambda$ under both AWGN model and Poisson noise model.

*Case I: Gaussian noise.* Let $\hat{\boldsymbol{x}}$ denote the output of the ZS-NCD denoiser, and consider the AWGN model $\boldsymbol{y} = \boldsymbol{x} + \boldsymbol{z}$ with $\boldsymbol{z} \sim \mathcal{N}(0, \sigma_z^2 I_n)$. Then,

$$\tfrac{1}{n}\mathbb{E}\big[\|\boldsymbol{y} - \hat{\boldsymbol{x}}\|_2^2\big] = \tfrac{1}{n}\mathbb{E}\big[\|\boldsymbol{y} - \boldsymbol{x} + \boldsymbol{x} - \hat{\boldsymbol{x}}\|_2^2\big] = \sigma_z^2 + \tfrac{1}{n}\|\boldsymbol{x} - \hat{\boldsymbol{x}}\|_2^2 + \tfrac{2}{n}\mathbb{E}\big[\boldsymbol{z}^T(\boldsymbol{x} - \hat{\boldsymbol{x}})\big], \tag{10}$$

the first term is the noise variance, and the second is the true denoising error. While $\boldsymbol{z}$ and $\hat{\boldsymbol{x}}$ are not fully independent, they are intuitively weakly correlated in successful denoising regimes, where the estimate $\hat{\boldsymbol{x}}$ depends only indirectly on the noise. Thus, the cross term is expected to be small: $\tfrac{1}{n}\mathbb{E}\big[\boldsymbol{z}^\top(\boldsymbol{x} - \hat{\boldsymbol{x}})\big] \approx 0$. This approximation suggests that, ideally, under low-noise regimes, $\tfrac{1}{n}\|\boldsymbol{y} - \hat{\boldsymbol{x}}\|_2^2$ is expected to be close to $\sigma_z^2$. Based on this insight, we propose a simple and effective heuristic for choosing $\lambda$: select $\lambda$ such that $\tfrac{1}{n}\|\boldsymbol{y} - \hat{\boldsymbol{x}}\|_2^2$ is closest to the known noise variance $\sigma_z^2$. This procedure can be implemented efficiently via a tree-based search strategy, as described in Algorithm 1. To apply Algorithm 1, one needs an estimate of the noise power $\sigma_z^2$. This is a well-studied problem and there exist robust algorithms for estimating the variance of noise [2, 53]. For example, in [2], it is shown that the noise power can be estimated from the median of the absolute differences of wavelet coefficients.

Finally, we observe that the performance of ZS-NCD is relatively robust to the choice of $\lambda$. For instance, on the Nuclei dataset (Figure 3), ZS-NCD outperforms the state-of-the-art zero-shot learning-

based denoiser, ZS-Noise2Noise, across a wide range of $\lambda$ values, for both $\sigma_z = 10$ and $\sigma_z = 20$. A similar approach can also be applied to the case of Poisson noise, as well.

*Case II: Poisson noise* In the case of Poisson noise, in addition to estimating $\lambda$, we need to estimate $\alpha$, which is used to normalize the measurements, in both the MSE-based and the MLE-based methods. Note that in the case of Poisson noise, $\mathbb{E}[y_i] = \alpha x_i$, and therefore, with high probability, $\frac{1}{n}\sum_{i=1}^{n} y_i \approx \alpha \frac{1}{n}\sum_{i=1}^{n} x_i$. Using this observation, and assuming that $\frac{1}{n}\sum_{i=1}^{n} x_i \approx 0.5$, we estimate $\alpha$ as $\hat{\alpha} = \frac{2}{n}\sum_{i=1}^{n} y_i$. We then use the estimated noise level $\hat{\alpha}$ to normalize both MSE and MLE based optimization for denoising Poisson noise. See Table 1 for the result of this noise parameter estimation on a sample image. Having an estimate of noise parameter $\alpha$, we then follow a similar procedure we used in the case of AWGN to set the parameter $\lambda$. Specifically, we write the MSE between normalized $\boldsymbol{y}$ and the denoised image $\hat{\boldsymbol{x}}$ as

$$\frac{1}{n}\mathbb{E}\big[\|\boldsymbol{y}/\alpha - \hat{\boldsymbol{x}}\|_2^2\big] = \frac{1}{n}\mathbb{E}\big[\|\boldsymbol{y}/\alpha - \boldsymbol{x} + \boldsymbol{x} - \hat{\boldsymbol{x}}\|_2^2\big]$$

$$= \frac{1}{n\alpha}\sum_{i=1}^{n} x_i + \frac{1}{n}\|\boldsymbol{x} - \hat{\boldsymbol{x}}\|_2^2 + \frac{2}{n}\mathbb{E}\big[(\boldsymbol{y} - \alpha\boldsymbol{x})^T(\boldsymbol{x} - \hat{\boldsymbol{x}})\big]. \qquad (11)$$

Again, assuming that we are in a low-noise regime, i.e., the second and the third terms in (11) are close to zero, and using $\frac{1}{n}\sum_{i=1}^{n} x_i \approx 0.5$ approximation, it follows that $\frac{1}{n}\mathbb{E}\big[\|\boldsymbol{y}/\alpha - \hat{\boldsymbol{x}}\|_2^2\big] \approx \frac{1}{2\alpha}$. This implies that, in the case of Poisson noise, we can still use Algorithm 1 to find $\lambda$, after updating $\|\hat{\boldsymbol{x}}^{(k)} - \boldsymbol{y}\|^2 > n\sigma_z^2$ to $\|\hat{\boldsymbol{x}}^{(k)} - \boldsymbol{y}\|^2 > \frac{1}{2\alpha}$. We empirically observe that MSE($\boldsymbol{y}/\alpha, \hat{\boldsymbol{x}}$) in training our networks is close to $\frac{1}{2\alpha}$ as reported in Table 1, which indicates that $\frac{1}{2\alpha}$ is a good approximation of the MSE that can be used as a threshold for selecting $\lambda$. When $\alpha$ is not known, we obtain the estimate $\hat{\alpha}$, and use $\frac{1}{2\hat{\alpha}}$ as a valid threshold to set $\lambda$.

Table 1: Analyzing the estimation of Poisson noise parameter for *Barbara* image in Set11 (MSE values are reported in terms of PSNR).

| true $\alpha$ | estimated $\hat{\alpha}$ | empirical MSE($\boldsymbol{y}/\alpha, \hat{\boldsymbol{x}}$) [dB] | $1/(2\alpha)$ [dB] | $1/(2\hat{\alpha})$ [dB] |
|---|---|---|---|---|
| 25 | 23.02 | 17.12 | 16.98 | 16.63 |
| 50 | 46.05 | 19.66 | 20.00 | 19.64 |

## 5   Experiments

In this section, we evaluate the denoising performance of ZS-NCD on both synthetic and real-world noise, across natural and microscopy images. We compare against representative zero-shot denoisers, including both traditional and learning-based methods. All baselines are dataset-free, i.e., they operate solely on the noisy image to be denoised. For non-learning methods, we include JPEG-2K and BM3D. Although rarely used as a denoising baseline, JPEG-2K provides a useful point of comparison from the perspective of compression-based denoising, as it represents a fixed, pre-defined compression code. For learning-based methods, we evaluate Deep Image Prior (DIP) [21], Deep Decoder (DD) [22], Zero-Shot Noise2Self (ZS-N2S) [12], Self2Self (S2S) [23], and Zero-Shot Noise2Noise (ZS-N2N) [24].

Due to instability in training for several baselines, we report their **best achieved performance** (with early stopping or model selection), whereas ZS-NCD is evaluated at its **final training iteration**, without manual tuning or stopping criteria.

**Natural images with synthetic noise**    We consider two synthetic noise models, AWGN $\mathcal{N}(0, \sigma_z^2)$, where $\sigma_z$ is the standard deviation of the Gaussian distribution, and Poisson noise defined as Poisson($\alpha x$), where $\alpha$ is the scale factor. Note that Poisson noise is signal-dependent noise with $\mathbb{E}[\boldsymbol{y}] = \alpha\boldsymbol{x}$. To re-scale the noisy image to the range of clean image, we followed the literature by assuming that the the scale $\alpha$ is known, and normalize the noisy image as $\boldsymbol{y}/\alpha$ in the experiments of this section. We evaluate on grayscale *Set11* [25], RGB *Set13* [54] (center-cropped to $192 \times 192$) and *Kodak24* [55] datasets. Table 2 presents the denoising performance of various methods. BM3D achieves the strongest results on grayscale images, though it relies on accurate knowledge of the noise power parameter. Existing learning-based zero-shot denoisers, in contrast, often exhibit inconsistent

Table 2: Denoising performance comparison under AWGN and Poisson Noise, average PSNR(dB) and SSIM are reported. Best results are in **bold**, second-best are underlined.

| Noise Parameter | | AWGN, $\mathcal{N}(0, \sigma^2)$ | | | Poisson, $\text{Poisson}(\alpha x)/\alpha$ | | |
|---|---|---|---|---|---|---|---|
| $\sigma$ or $\alpha$ | Method | Set11 | Set13 | Kodak24 | Set11 | Set13 | Kodak24 |
| | JPEG2K | 27.45 / 0.7699 | 26.69 / 0.7543 | 27.86 / 0.7457 | 22.35 / 0.5882 | 21.76 / 0.5494 | 22.56 / 0.5249 |
| | BM3D | **32.22 / 0.8991** | 31.15 / 0.8808 | 32.37 / 0.8754 | **26.66 / 0.7505** | 25.64 / 0.6912 | 27.04 / 0.6900 |
| | DIP | 29.11 / 0.7990 | 30.31 / 0.8570 | 31.42 / 0.8454 | 23.69 / 0.5863 | 25.14 / 0.6916 | 26.37 / 0.6761 |
| 15 | DD | 28.83 / 0.8215 | 29.22 / 0.8371 | 28.71 / 0.8016 | 24.37 / 0.6629 | 24.96 / 0.7006 | 25.59 / 0.6679 |
| | S2S | 26.81 / 0.8158 | 20.61 / 0.6879 | 23.08 / 0.7695 | 21.75 / 0.6872 | 19.23 / 0.6553 | 22.52 / 0.7418 |
| | ZS-N2S | 28.92 / 0.8495 | 18.18 / 0.5690 | 18.68 / 0.5540 | 25.06 / 0.7051 | 21.23 / 0.6066 | 22.24 / 0.6170 |
| | ZS-N2N | 30.01 / 0.8169 | 30.95 / 0.8701 | 32.30 / 0.8650 | 24.04 / 0.5766 | 25.37 / 0.6878 | 26.80 / 0.6757 |
| | **ZS-NCD** | 31.35 / 0.8580 | **31.93 / 0.8983** | **33.18 / 0.9026** | 25.65 / 0.7132 | **26.44 / 0.7434** | **27.64 / 0.7432** |
| | JPEG2K | 24.91 / 0.6997 | 24.32 / 0.6676 | 25.43 / 0.6550 | 23.03 / 0.6108 | 22.65 / 0.5952 | 23.58 / 0.5680 |
| | BM3D | **29.79 / 0.8523** | 28.81 / 0.8213 | 29.98 / 0.8092 | 22.70 / 0.5741 | 22.17 / 0.5992 | 24.13 / 0.5931 |
| | DIP | 26.60 / 0.7128 | 27.85 / 0.7837 | 28.90 / 0.7738 | 24.94 / 0.6512 | 26.13 / 0.7289 | 27.49 / 0.7243 |
| 25 | DD | 26.93 / 0.7530 | 27.40 / 0.7832 | 27.62 / 0.7496 | 25.48 / 0.7022 | 26.04 / 0.7373 | 26.56 / 0.7060 |
| | S2S | 23.32 / 0.7306 | 17.95 / 0.5998 | 20.69 / 0.6949 | 23.40 / 0.7355 | 20.18 / 0.6927 | 23.09 / 0.7674 |
| | ZS-N2S | 27.30 / 0.7971 | 20.39 / 0.6200 | 20.89 / 0.6156 | 26.01 / 0.7478 | 21.19 / 0.6312 | 21.47 / 0.6277 |
| | ZS-N2N | 27.18 / 0.7173 | 28.36 / 0.8001 | 29.54 / 0.7798 | 25.40 / 0.6432 | 26.75 / 0.7455 | 28.21 / 0.7374 |
| | **ZS-NCD** | 28.93 / 0.8079 | **29.33 / 0.8351** | **30.60 / 0.8144** | **27.10 / 0.7431** | **27.60 / 0.7827** | **28.77 / 0.7677** |
| | JPEG2K | 22.05 / 0.5794 | 21.43 / 0.5295 | 22.17 / 0.5055 | 24.77 / 0.6811 | 24.25 / 0.6696 | 25.52 / 0.6608 |
| | BM3D | **26.56 / 0.7619** | 25.78 / 0.7134 | 27.06 / 0.7047 | 23.09 / 0.5787 | 23.00 / 0.6281 | 24.49 / 0.6008 |
| | DIP | 23.46 / 0.5783 | 24.82 / 0.6748 | 25.90 / 0.6494 | 26.30 / 0.7004 | 27.72 / 0.7845 | 29.12 / 0.7845 |
| 50 | DD | 24.01 / 0.6584 | 24.56 / 0.6779 | 24.98 / 0.6413 | 26.87 / 0.7455 | 27.43 / 0.7867 | 27.71 / 0.7543 |
| | S2S | 17.41 / 0.5200 | 14.21 / 0.3938 | 17.00 / 0.5325 | 25.70 / 0.7896 | 21.75 / 0.7365 | 23.88 / 0.8014 |
| | ZS-N2S | 24.74 / 0.6883 | 20.62 / 0.5880 | 20.05 / 0.5774 | 27.08 / 0.7855 | 20.75 / 0.6033 | 20.25 / 0.5993 |
| | ZS-N2N | 23.52 / 0.5457 | 24.67 / 0.6444 | 25.82 / 0.6151 | 27.26 / 0.7216 | 28.57 / 0.8112 | 30.13 / 0.8076 |
| | **ZS-NCD** | 25.58 / 0.7144 | **25.87 / 0.7269** | **27.89 / 0.7464** | **28.44 / 0.7914** | **29.09 / 0.8223** | **30.60 / 0.8235** |

performance across noise levels and image resolutions. For example, ZS-N2S and Self2Self degrade on high-resolution images, likely due to the limitations of training with masked pixels. ZS-N2N performs well on high-resolution images from Kodak24 but suffers on lower-resolution images in Set13 ($192 \times 192$), as it is trained to map between two downscaled versions of the same noisy image. In comparison, ZS-NCD maintains robust performance across different noise levels and image sizes. The more realistic case of not having access to the noise parameter $\alpha$ was discussed in Section 4. In both noise regimes, we use MSE as the loss function. However, for Poisson noise, minimizing the negative log-likelihood is also a natural choice. We defer the results using this loss to Appendix B.3.

Table 3: First 2 rows: Denoising performance (average PSNR / SSIM of 6 images) under AWGN $\mathcal{N}(0, \sigma^2 I)$ on Mouse Nucle fluorescence microscopy images (image size $128 \times 128$). Noise levels are 10 and 20. Last row: Real camera denoising performance on camera image dataset: PolyU. The images are cropped into size of $512 \times 512$. We report the average PSNR / SSIM of 6 random images.

| $\sigma$ | JPEG2K | BM3D | DIP | DD | ZS-N2N | ZS-N2S | S2S | ZS-NCD |
|---|---|---|---|---|---|---|---|---|
| 10 | 32.89 / 0.8294 | **38.65 / 0.9640** | 36.43 / 0.8789 | 37.33 / 0.9533 | 36.17 / 0.9319 | 31.26 / 0.8812 | 12.63 / 0.2966 | 38.23 / 0.9508 |
| 20 | 28.57 / 0.6986 | **34.96 / 0.9296** | 32.32 / 0.7889 | 33.50 / 0.9092 | 32.25 / 0.8532 | 30.41 / 0.8600 | 10.09 / 0.1559 | 34.71 / 0.9093 |
| Unknown | 32.89 / 0.8294 | 35.71 / 0.9506 | 35.43 / 0.9408 | 34.83 / 0.9395 | 34.07 / 0.9028 | 23.61 / 0.8344 | 35.66 / 0.9527 | **35.84 / 0.9534** |

**Fluorescence microscopy and real camera images** To evaluate performance in low-data and domain-shift settings, we test ZS-NCD on Mouse Nuclei fluorescence microscopy images [56], which differ significantly from natural images in structure and texture. We also assess real-world denoising using the PolyU dataset [57], which contains high-resolution images captured by Canon, Nikon, and Sony cameras. Ground-truth images are obtained by averaging multiple captures, while the noisy inputs are single-shot acquisitions. Results are shown in Table 3. ZS-NCD consistently outperforms other learning-based zero-shot denoisers, demonstrating robustness to unknown noise models and non-natural image distributions.

Table 4: Comparison of AWGN denoising of Conv and MLP based ZS-NCD on Set11.

| ZS-NCD | $\sigma = 25$ | $\sigma = 50$ |
|---|---|---|
| Conv | 28.93 / 0.8079 | 25.58 / 0.7144 |
| MLP | 29.52 / 0.8363 | 25.89 / 0.7306 |

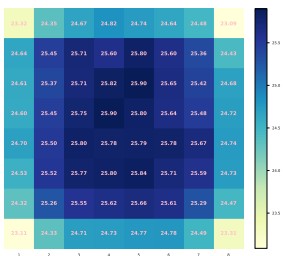

Figure 4: Denoising *Parrot* using ZS-NCD, where only a single pixel from each overlapping patch (stride 1) is retained after compression. (AWGN, $\sigma_z = 25$.) Each heatmap value indicates the PSNR achieved when denoising is based solely on the pixel at that specific location within each patch.

**Robustness to overfitting.** Most learning-based zero-shot methods are prone to overfitting due to the lack of clean targets and the use of overparameterized networks. In contrast, ZS-NCD, grounded in compression-based denoising theory, overcomes this issue given the entropy constraint. To further highlight this key aspect of ZS-NCD, we replace the convolutional encoder-decoder ($\approx 0.4$M params) with a fully connected MLP ($\approx 2.3$M params) and observe that, instead of degradation, the performance improves using the same $\lambda$ (see Table 4).

**Effect of overlapping patch aggregation.** As described in Section 4 and illustrated in Fig. 1, ZS-NCD denoises each pixel by aggregating outputs from overlapping patches, where each patch is first compressed and then decompressed using a learned neural compression model. Intuitively, one might expect the most accurate reconstruction for a given pixel to come from the patch in which it lies at the center, as this location benefits from the largest available spatial context, which has been observed in [58].

This observation leads to the question: *Does averaging over overlapping reconstructions improve denoising quality, or would it suffice to use only the patch where pixel appears at a fixed position (e.g., the center)?* From a computational perspective, both strategies are equivalent, since in both methods every patch is processed, but in averaging scheme, each patch contributes to all the pixels it covers.

To investigate this, we conducted an ablation in which, instead of averaging, each pixel $(i, j)$ is reconstructed solely from one of the $k \times k$ patches in which it appears, using a fixed location in the patch (e.g., top-left, center, etc.). The results are shown in Fig. 4, where each heatmap entry reports the PSNR obtained by using only that specific location in the patch for reconstruction. As expected, performance is best when the pixel is centrally located, and degrades as it moves toward the patch boundaries.

However, the key observation is that averaging across all overlapping reconstructions yields a substantial performance gain. For instance, in denoising Parrot (from Set11 dataset), the best single-location reconstruction achieves 25.90 dB (center), while averaging achieves 28.14 dB, a gain of over 2 dB. This highlights the denoising benefit of combining multiple noisy views of each pixel, consistent with principles from ensembling and variance reduction.

# 6 Conclusions

We have studied maximum likelihood compression-based denoising, and provided theoretical characterization of its performance under both AWGN and Poisson noise. Furthermore, we introduced ZS-NCD, a new zero-shot neural-compression-based denoising and demonstrated that it achieves state-of-the-art performance among zero-shot methods, in both AWGN and Poisson denoising.

The presented theoretical results are derived by assuming a fixed (e.g., pre-trained/defined) compression code. Extending these results to the case of zero-shot learned compression codes is an interesting direction for future research.

# Acknowledgment

A.Z., X.C., S.J. were supported by NSF CCF-2237538.

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

# A Proofs

## A.1 Auxiliary lemmas

Before stating the proofs of the mains theorems, here we state some lemmas that will be used later in the proofs.

**Lemma 1.** *Assume that $0 < \alpha_m \le \alpha_1, \alpha_2 \le \alpha_M < \infty$. Then,*

$$(\frac{\alpha_m}{\alpha_M})^2 \frac{(\alpha_2 - \alpha_1)^2}{2\alpha_1} \le D_{KL}(\text{Poisson}(\alpha_1)\|\text{Poisson}(\alpha_2)) \le (\frac{\alpha_M}{\alpha_m})^2 \frac{(\alpha_2 - \alpha_1)^2}{2\alpha_1}.$$

**Lemma 2.** *Consider independent Poisson random variables $Y_1, \ldots, Y_n$, where $Y_i \sim \text{Poisson}(\alpha_i)$. Consider $w_1, \ldots, w_n \in \mathbb{R}$. Let $\sigma_n^2 = \sum_{i=1}^n w_i^2 \alpha_i$ and $w_M \triangleq \max_{i \in \{1,\ldots,n\}} |w_i|$. Then, for any $t \in [0, \frac{3\sigma_n^2}{2w_M}]$,*

$$\mathbb{P}\left( \sum_{i=1}^n w_i Y_i \ge \sum_{i=1}^n w_i \alpha_i + t \right) \le \exp(-\frac{t^2}{2(\sigma_n^2 + w_M t/3)}). \tag{12}$$

*and*

$$\mathbb{P}\left( \sum_{i=1}^n w_i Y_i \le \sum_{i=1}^n w_i \alpha_i - t \right) \le \exp(-\frac{t^2}{2(\sigma_n^2 + w_M t/3)}). \tag{13}$$

### A.1.1 Proof of Lemma 1

*Proof.*

$$D_{\text{KL}}(\text{Poisson}(\alpha_1)\|\text{Poisson}(\alpha_2)) = \alpha_2 - \alpha_1 + \alpha_1 \log \frac{\alpha_1}{\alpha_2} = \alpha_2 - \alpha_1 - \alpha_1 \log(1 + \frac{\alpha_2 - \alpha_1}{\alpha_1}). \tag{14}$$

Using the Taylor's theorem,

$$\log(1 + u) = u - f''(\alpha)\frac{u^2}{2}, \tag{15}$$

where $f(u) = \log(1 + u)$ and $\alpha \in (0, u)$. Note that

$$f''(u) = -\frac{1}{(1 + u)^2}.$$

Letting $u = \frac{\alpha_2 - \alpha_1}{\alpha_1}$, for $\alpha \in (0, u)$,

$$-(\frac{\alpha_M}{\alpha_m})^2 \le f''(\alpha) \le -(\frac{\alpha_m}{\alpha_M})^2. \tag{16}$$

Combining (14), (15) and (16) yields the desired result.

$\square$

### A.1.2 Proof of Lemma 2

*Proof.* Define

$$\mu_n = \mathbb{E}\left[ \sum_{i=1}^n w_i Y_i \right] = \sum_{i=1}^n w_i \alpha_i,$$

and

$$\sigma_n^2 = \mathbb{E}\left[ (\sum_{i=1}^n w_i(Y_i - \alpha_i))^2 \right] = \sum_{i=1}^n w_i^2 \alpha_i.$$

Consider $s > 0$, then using the Chernoff bound, we have

$$\mathbb{P}\left(\sum_{i=1}^{n} w_i Y_i \geq \sum_{i=1}^{n} w_i \alpha_i + t\right) \leq \frac{\prod_{i=1}^{n} \mathbb{E}[\exp(sw_i(Y_i - \alpha_i))]}{\exp(st)}$$

$$= \exp(\sum_{i=1}^{n}(\alpha_i(e^{sw_i} - 1 - sw_i)) - st). \tag{17}$$

Note that for $u \in (-1, 1)$, $e^u - 1 - u \leq \frac{u^2}{2(1-u/3)}$. Assuming that $s \leq \frac{1}{w_M}$, then $|sw_i| < 1$, for all $i$. Therefore,

$$\exp(\sum_{i=1}^{n}(\alpha_i(e^{sw_i} - 1 - sw_i)) - st) \leq \exp(\sum_{i=1}^{n} \alpha_i(\frac{(sw_i)^2}{2(1 - sw_i/3)}) - st)$$

$$\leq \exp(\sum_{i=1}^{n} \alpha_i(\frac{(sw_i)^2}{2(1 - sw_M/3)}) - st)$$

$$= \exp(\frac{s^2\sigma_n^2}{2(1 - sw_M/3)} - st). \tag{18}$$

Evaluating this bound at $s = \frac{t}{\sigma_n^2 + w_M t/3}$, since $1 - sw_M/3 = \frac{\sigma_n^2}{\sigma_n^2 + w_M t/3}$, it follows that

$$\frac{s^2\sigma_n^2}{2(1 - sw_M/3)} - st = -\frac{st}{2} = -\frac{t^2}{2(\sigma_n^2 + w_M t/3)}. \tag{19}$$

To derive the other bound, we can follow the same steps and apply Chernoff bound as done in (17) to get

$$\mathbb{P}\left(\sum_{i=1}^{n} w_i Y_i \leq \mu_n - t\right) \leq \exp(\sum_{i=1}^{n} \alpha_i(e^{-sw_i} - 1 + sw_i) - st). \tag{20}$$

We now use the inequality for $u \in (-1, 1)$:

$$e^{-u} - 1 + u \leq \frac{u^2}{2(1 + u/3)}.$$

Assume $s \leq \frac{1}{w_M}$ so that $s|w_i| \leq 1$ for all $i$. Then:

$$\sum_{i=1}^{n} \alpha_i(e^{-sw_i} - 1 + sw_i) \leq \sum_{i=1}^{n} \alpha_i \cdot \frac{(sw_i)^2}{2(1 + sw_i/3)}$$

$$\leq \sum_{i=1}^{n} \alpha_i \cdot \frac{(sw_i)^2}{4/3}$$

$$= \frac{s^2\sigma_n^2}{4/3}. \tag{21}$$

Hence,

$$\mathbb{P}\left(\sum_{i=1}^{n} w_i Y_i \leq \mu_n - t\right) \leq \exp\left(\frac{s^2\sigma_n^2}{4/3} - st\right). \tag{22}$$

Setting $s = \frac{3t}{2\sigma_n^2}$, which satisfies $s \leq \frac{1}{w_M}$,

$$\frac{s^2\sigma_n^2}{4/3} - st = -\frac{3t^2}{4\sigma_n^2} \leq -\frac{t^2}{2(\sigma_n^2 + w_M t/3)}. \tag{23}$$

$\square$

## A.2 Proof of Theorem 1

*Proof.* Recall that $y = x + z$, with $z$ is i.i.d. $\mathcal{N}(0, \sigma_z^2)$, and

$$\hat{x} = \arg\min_{\mathbf{c} \in \mathcal{C}} \|\mathbf{c} - y\|_2^2, \qquad \tilde{x} = \arg\min_{\mathbf{c} \in \mathcal{C}} \|\mathbf{c} - x\|_2^2.$$

Since both $\hat{x}, \tilde{x}$ are in $\mathcal{C}$,

$$\|\hat{x} - y\|_2^2 \le \|\tilde{x} - y\|_2^2 \tag{24}$$

$$\|(\hat{x} - x) - z\|_2^2 \le \|(\tilde{x} - x) - z\|_2^2 \tag{25}$$

$$\|\hat{x} - x\|^2 - 2\langle z, \hat{x} - x\rangle + \|z\|^2 \le \|\tilde{x} - x\|^2 - 2\langle z, \tilde{x} - x\rangle + \|z\|^2 \tag{26}$$

$$\|\hat{x} - x\|^2 \le \|\tilde{x} - x\|^2 - 2\langle z, \tilde{x} - x\rangle + 2\langle z, \hat{x} - x\rangle \tag{27}$$

$$\|\hat{x} - x\|^2 \le \|\tilde{x} - x\|^2 + 2|\langle z, \tilde{x} - x\rangle| + 2|\langle z, \hat{x} - x\rangle|. \tag{28}$$

Let $\mathbf{e} = \hat{x} - x$ denote the error of the compression-based estimate of ground truth $x$ from its noisy version $y$, and $d = \tilde{x} - x$ denote the distortion from the compressing the ground truth $x$ with the compression code $\mathcal{C}$, then we have

$$\|\mathbf{e}\|^2 \le \|d\|^2 + 2|\langle z, \mathbf{e}\rangle| + 2|\langle z, d\rangle| \tag{29}$$

$$= \|d\|^2 + 2\|\mathbf{e}\| \left|\langle z, \frac{\mathbf{e}}{\|\mathbf{e}\|}\rangle\right| + 2\|d\| \left|\langle z, \frac{d}{\|d\|}\rangle\right|. \tag{30}$$

For any possible reconstruction $\mathbf{c} \in \mathcal{C}$, we define error vector $\mathbf{e}^{(\mathbf{c})} = \mathbf{c} - x$. Given $t_1, t_2 > 0$, define event $\mathcal{E}_1$ and $\mathcal{E}_2$ as

$$\mathcal{E}_1 = \left\{ \left|\sum_{i=1}^{n} z_i \frac{e_i^{(\mathbf{c})}}{\|\mathbf{e}^{(\mathbf{c})}\|}\right| \le t_1 : \mathbf{c} \in \mathcal{C} \right\}. \tag{31}$$

and

$$\mathcal{E}_2 = \left\{ \left|\sum_{i=1}^{n} z_i \frac{e_i^{(\tilde{x})}}{\|\mathbf{e}^{(\tilde{x})}\|}\right| \le t_2 \right\}, \tag{32}$$

respectively. Conditioned on $\mathcal{E}_1 \cap \mathcal{E}_2$, it follows from (30) that

$$\|\mathbf{e}\|^2 \le \|d\|^2 + 2t_1\|\mathbf{e}\| + 2t_2\|d\|. \tag{33}$$

Therefore,

$$\|\mathbf{e}\|^2 - 2t_1\|\mathbf{e}\| + t_1^2 \le \|d\|^2 + 2t_2\|d\| + t_2^2 + (t_1^2 - t_2^2), \tag{34}$$

$$\left|\|\mathbf{e}\| - t_1\right| \le \sqrt{(\|d\| + t_2)^2 + (t_1^2 - t_2^2)}, \tag{35}$$

and finally,

$$\|\mathbf{e}\|_2 \le \|d\|_2 + t_1 + t_2 + \sqrt{t_1^2 - t_2^2}, \tag{36}$$

where the last line follows because $\sqrt{a + b} \le \sqrt{a} + \sqrt{b}$, for all $a, b > 0$. To finish the proof we need to bound $\mathbb{P}((\mathcal{E}_1 \cap \mathcal{E}_2)^c)$ and set parameters $t_1$ and $t_2$.

Note that for each $\mathbf{c}$, $\frac{\mathbf{e}^{(\mathbf{c})}}{\|\mathbf{e}^{(\mathbf{c})}\|}$ is a unit vector in $\mathbb{R}^n$. Therefore, $\sum_{i=1}^{n} z_i \frac{e_i^{(\mathbf{c})}}{\|\mathbf{e}^{(\mathbf{c})}\|} \sim \mathcal{N}(0, \sigma_z^2)$. Hence,

$$\mathbb{P}\left(\left|\sum_{i=1}^{n} z_i \frac{e_i^{(c)}}{\|\mathbf{e}^{(c)}\|}\right| \ge t\right) \le 2\exp\left(-\frac{t^2}{2\sigma_z^2}\right). \tag{37}$$

Therefore, applying the union bound and noting that $|\mathcal{C}| \le 2^R$,

$$\mathbb{P}(\mathcal{E}_1^c) \le 2^{R+1} \exp\left(-\frac{t_1^2}{2\sigma_z^2}\right), \tag{38}$$

and

$$\mathbb{P}(\mathcal{E}_2^c) \leq 2 \exp\left(-\frac{t_2^2}{2\sigma_z^2}\right). \tag{39}$$

For $\eta \in (0, 1)$, set

$$t_1 = \sigma_z \sqrt{2 \ln 2R(1 + \eta)},$$

and

$$t_2 = \sigma_z \sqrt{2 \ln 2R\eta}.$$

Then,

$$\mathbb{P}(\mathcal{E}_1^c \cup \mathcal{E}_2^c) \leq 2^{-\eta R + 2}. \tag{40}$$

Using the selected values of $t_1$ and $t_2$ in (36) yields the desired result, i.e.,

$$\frac{1}{\sqrt{n}} \|\boldsymbol{x} - \hat{\boldsymbol{x}}\|_2 \leq \sqrt{\delta} + 2\sigma_z \sqrt{\frac{(2 \ln 2)R}{n}} (1 + 2\sqrt{\eta}),$$

where we have used the fact that $\frac{1}{n} \|\boldsymbol{d}\|_2^2 \leq \delta$. $\qquad\square$

### A.3 Proof of Corollary 1

*Proof.* We start by designing a lossy compression code for the set of signals in $\mathcal{Q}_n$, defined as

$$\mathcal{Q}_n = \{\boldsymbol{x} \in \mathbb{R}^n : \|\boldsymbol{x}\|_0 \leq k, \|\boldsymbol{x}\|_\infty \leq 1\}. \tag{41}$$

For a $k$-sparse $\boldsymbol{x} \in \mathcal{Q}_n$, let $\boldsymbol{x}^{(k)} \in \mathbb{R}^k$ denote the $k$-dimensional vector derived from the non-zero coordinates of $\boldsymbol{x}$. We define a lossy compression code $(f, g)$ that achieves distortion $\delta$. Specifically, given a $k$-sparse $\boldsymbol{x} \in \mathcal{Q}_n$, the encoder operates as follows:

1. Encode the number of non-zero entries and their locations. This requires at most

$$\log_2 k + \log_2 \binom{n}{k} \leq \log_2 k + k \log_2(\frac{n\mathrm{e}}{k})$$

   bits.

2. Quantize the values of the non-zero coordinates such that overall distortion $\delta$ is achieved. To achieve this goal it quantizes each non-zero coordinate of $\boldsymbol{x}$ into $b$ bits. Let $[x_i]_b$ denote the $b$-bit quantized version of $x_i$. Then, $|x_i - [x_i]_b| \leq 2^{-b}$. Therefore, the overall $\ell_2$ distortion can be bounded as

$$\|\boldsymbol{x} - g(f(\boldsymbol{x}))\|_2^2 \leq k 2^{-2b}.$$

   Choosing

$$b = \lceil \frac{1}{2} \log_2 \frac{k}{n\delta} \rceil$$

   ensures that $\|\boldsymbol{x} - g(f(\boldsymbol{x}))\|_2^2 \leq n\delta$.

It can be observed that overall the number of bits required for describing the signals in $\mathcal{Q}_n$ within distortion $\delta$ can be bounded as

$$R \leq \frac{k}{2} \log_2(\frac{k}{n\delta}) + k \log_2 \frac{n}{k} + c_k,$$

where $c_k = \log_2 k + k(\log_2 \mathrm{e} + 1)$. Using the defined lossy compression code to solve (1) and applying Theorem 1, it follows that, with a probability larger than $1 - 2^{-\eta R + 2}$,

$$\frac{1}{\sqrt{n}} \|\boldsymbol{x} - \hat{\boldsymbol{x}}\|_2 \leq \sqrt{\delta} + \sigma_z C \sqrt{\frac{k}{2n} \log_2(\frac{k}{n\delta}) + \frac{k}{n} \log_2(\frac{n}{k}) + \frac{c_k}{n}}$$

$$= \sqrt{\delta} + \sigma_z C \sqrt{\frac{k}{2n} \log_2(\frac{n}{k\delta}) + \frac{c_k}{n}},$$

where $C = 2(1 + 2\sqrt{\eta})\sqrt{2 \ln 2}$. Let

$$\delta = \frac{1}{n}.$$

Then,

$$\frac{1}{\sqrt{n}}\|\boldsymbol{x} - \hat{\boldsymbol{x}}\|_2 \leq \frac{1}{\sqrt{n}} + \sigma_z C \sqrt{\frac{k \log_2 k}{2n} + \frac{k}{n} \log_2(\frac{n}{k}) + \frac{c_k}{n}}$$

$$= \frac{1}{\sqrt{n}} + \sigma_z C \sqrt{\frac{k}{n} \log_2(\frac{n}{k}) + \gamma_n},$$

where

$$\gamma_n = \frac{k \log_2 k}{2n} + \frac{c_k}{n}.$$

Finally, since $R \geq k \log_2(\frac{n}{k})$,

$$1 - 2^{-\eta R + 2} \geq 1 - \mathrm{e}^{-\eta k \log(\frac{n}{k})}.$$

$\square$

### A.4  Proof of Theorem 2

*Proof.* Recall that $\arg\min_{\mathbf{c} \in \mathcal{C}} \mathcal{L}(\mathbf{c}; \boldsymbol{y})$. Let

$$\tilde{\boldsymbol{x}} = \arg\min_{\mathbf{c} \in \mathcal{C}} \|\boldsymbol{x} - \mathbf{c}\|_2.$$

Since both $\hat{\boldsymbol{x}}$ and $\tilde{\boldsymbol{x}}$ are in $\mathcal{C}$, we have $\mathcal{L}(\hat{\boldsymbol{x}}; \boldsymbol{y}) \leq \mathcal{L}(\tilde{\boldsymbol{x}}; \boldsymbol{y})$, or

$$\sum_{i=1}^{n} (\alpha\hat{x}_i - y_i \log \hat{x}_i) \leq \sum_{i=1}^{n} (\alpha\tilde{x}_i - y_i \log \tilde{x}_i). \tag{42}$$

Given the input signal $\boldsymbol{x} \in \mathbb{R}^n$ and $\mathbf{c} \in \mathcal{C}$, let $\mathrm{Poisson}(\alpha\boldsymbol{x})$ and $\mathrm{Poisson}(\alpha\mathbf{c})$ denote the distributions corresponding to independent Poisson random variables with respective means $\alpha x_i$ and $\alpha c_i$. Note that

$$D_{\mathrm{KL}}(\mathrm{Poisson}(\alpha\boldsymbol{x})\|\mathrm{Poisson}(\alpha\mathbf{c})) = \sum_{i=1}^{n} \left(\alpha(c_i - x_i) + \alpha x_i \log \frac{x_i}{c_i}\right). \tag{43}$$

Adding $\sum_i(-\alpha x_i + \alpha x_i \log x_i)$ to the both sides of (42), it follows that

$$\sum_{i=1}^{n} (\alpha(\hat{x}_i - x_i) - y_i \log \hat{x}_i + \alpha x_i \log x_i) \leq \sum_{i=1}^{n} (\alpha(\tilde{x}_i - x_i) - y_i \log \tilde{x}_i + \alpha x_i \log x_i), \tag{44}$$

or

$$D_{\mathrm{KL}}(\mathrm{Poisson}(\alpha\boldsymbol{x})\|\mathrm{Poisson}(\alpha\hat{\boldsymbol{x}})) + \sum_{i=1}^{n}(\alpha x_i - y_i) \log \hat{x}_i$$

$$\leq D_{\mathrm{KL}}(\mathrm{Poisson}(\alpha\boldsymbol{x})\|\mathrm{Poisson}(\alpha\tilde{\boldsymbol{x}})) + \sum_{i=1}^{n}(\alpha x_i - y_i) \log \tilde{x}_i. \tag{45}$$

Given $t_1, t_2 > 0$, define events $\mathcal{E}_1$ and $\mathcal{E}_2$ as

$$\mathcal{E}_1 = \left\{\sum_{i=1}^{n}(y_i - \alpha x_i) \log c_i \leq t_1 : \forall\mathbf{c} \in \mathcal{C}\right\} \tag{46}$$

and

$$\mathcal{E}_2 = \left\{\sum_{i=1}^{n}(y_i - \alpha x_i) \log \tilde{x}_i \geq -t_2\right\}, \tag{47}$$

respectively. Conditioned on $\mathcal{E}_1 \cap \mathcal{E}_2$,

$$D_{\text{KL}}(\text{Poisson}(\alpha \boldsymbol{x}) \| \text{Poisson}(\alpha \hat{\boldsymbol{x}})) \leq D_{\text{KL}}(\text{Poisson}(\alpha \boldsymbol{x}) \| \text{Poisson}(\alpha \tilde{\boldsymbol{x}})) + t_1 + t_2, \quad (48)$$

and consequently from Lemma 1,

$$\frac{x_{\min}^2}{x_{\max}^3} \alpha \| \boldsymbol{x} - \hat{\boldsymbol{x}} \|_2^2 \leq \frac{x_{\max}^2}{x_{\min}^3} \alpha \| \boldsymbol{x} - \tilde{\boldsymbol{x}} \|_2^2 + (t_1 + t_2). \quad (49)$$

To finish the proof, we bound $\mathbb{P}(\mathcal{E}_1 \cap \mathcal{E}_2)$ and set $t_1$ and $t_2$.

To bound $\mathbb{P}(\mathcal{E}_1^c)$, we apply Lemma 2, where for each $\mathbf{c}$, we set $w_i(\mathbf{c}) = \log \frac{1}{c_i}$. Then,

$$\sigma^n(\mathbf{c}) = \sum_{i=1}^{n} (\log \frac{1}{c_i})^2 \alpha x_i \leq \alpha \beta^2 \sum_{i=1}^{n} x_i,$$

and

$$w_M = \max_i |w_i| \leq \beta,$$

where

$$\beta = \log(\frac{1}{x_{\min}}).$$

Therefore, using the union bound, it follows that

$$\mathbb{P}(\mathcal{E}_1^c) \leq 2^R \exp(-\frac{t_1^2}{2(\alpha \beta^2 \sum_{i=1}^{n} x_i + \beta t_1 / 3)})$$

$$\leq 2^R \exp(-\frac{t_1^2}{2(n \alpha \beta^2 x_{\max} + \beta t_1 / 3)}). \quad (50)$$

To bound $\mathbb{P}(\mathcal{E}_2^c)$, we again apply Lemma 2, with $w_i = \log \frac{1}{\tilde{x}_i}$, and derive

$$\mathbb{P}(\mathcal{E}_2^c) \leq \exp(-\frac{t_2^2}{2(\alpha \beta^2 \sum_{i=1}^{n} x_i + \beta t_2 / 3)})$$

$$\leq \exp(-\frac{t_2^2}{2(n \alpha \beta^2 x_{\max} + \beta t_1 / 3)}). \quad (51)$$

Setting $t_1$ and $t_2$ such that they are both smaller than $3n\alpha\beta$, and noting that $x_{\max} < 1$, we have

$$\mathbb{P}(\mathcal{E}_1^c \cap \mathcal{E}_2^c) \leq 2^R \exp(-\frac{t_1^2}{4n\alpha\beta^2}) + \exp(-\frac{t_2^2}{4n\alpha\beta^2}). \quad (52)$$

Choosing $t_1 = \beta \sqrt{\frac{4}{\ln 2} nR(1+\eta)\alpha}$ and $t_2 = \beta \sqrt{\frac{4}{\ln 2} nR\eta\alpha}$, it follows that

$$\mathbb{P}(\mathcal{E}_1^c \cap \mathcal{E}_2^c) \leq 2^{-\eta R}. \quad (53)$$

$$\square$$

## A.5 Proof of Theorem 3

*Proof.* Recall that

$$\hat{\boldsymbol{x}} = \arg\min_{\mathbf{c} \in \mathcal{C}} \| \mathbf{c} - \boldsymbol{y}/\alpha \|_2^2, \qquad \tilde{\boldsymbol{x}} = \arg\min_{\mathbf{c} \in \mathcal{C}} \| \mathbf{c} - \boldsymbol{x} \|_2^2.$$

Following the similar setup as in Section A.2, we get

$$\| \hat{\boldsymbol{x}} - \boldsymbol{y}/\alpha \|^2 \leq \| \tilde{\boldsymbol{x}} - \boldsymbol{y}/\alpha \|^2$$

$$\| \hat{\boldsymbol{x}} - \boldsymbol{x} + \boldsymbol{x} - \boldsymbol{y}/\alpha \|^2 \leq \| \tilde{\boldsymbol{x}} - \boldsymbol{x} + \boldsymbol{x} - \boldsymbol{y}/\alpha \|^2$$

$$\| \hat{\boldsymbol{x}} - \boldsymbol{x} \|^2 - 2\langle \boldsymbol{y}/\alpha - \boldsymbol{x}, \hat{\boldsymbol{x}} - \boldsymbol{x} \rangle + \| \boldsymbol{y}/\alpha - \boldsymbol{x} \|^2 \leq \| \tilde{\boldsymbol{x}} - \boldsymbol{x} \|^2 - 2\langle \boldsymbol{y}/\alpha - \boldsymbol{x}, \tilde{\boldsymbol{x}} - \boldsymbol{x} \rangle + \| \boldsymbol{y}/\alpha - \boldsymbol{x} \|^2$$

$$\| \hat{\boldsymbol{x}} - \boldsymbol{x} \|^2 \leq \| \tilde{\boldsymbol{x}} - \boldsymbol{x} \|^2 - 2\langle \boldsymbol{y}/\alpha - \boldsymbol{x}, \tilde{\boldsymbol{x}} - \boldsymbol{x} \rangle + 2\langle \boldsymbol{y}/\alpha - \boldsymbol{x}, \hat{\boldsymbol{x}} - \boldsymbol{x} \rangle$$

$$\| \hat{\boldsymbol{x}} - \boldsymbol{x} \|^2 \leq \| \tilde{\boldsymbol{x}} - \boldsymbol{x} \|^2 + 2|\langle \boldsymbol{y}/\alpha - \boldsymbol{x}, \tilde{\boldsymbol{x}} - \boldsymbol{x} \rangle| + 2|\langle \boldsymbol{y}/\alpha - \boldsymbol{x}, \hat{\boldsymbol{x}} - \boldsymbol{x} \rangle|.$$

Defining e, $\boldsymbol{d}$ and $\mathbf{e}^{(\mathbf{c})}$, $\mathbf{c} \in \mathcal{C}$, as done in the proof of Theorem 1, we have

$$\|\mathbf{e}\|^2 \leq \|\boldsymbol{d}\|^2 + 2|\langle \boldsymbol{y}/\alpha - \boldsymbol{x}, \mathbf{e}\rangle| + 2|\langle \boldsymbol{y}/\alpha - \boldsymbol{x}, \boldsymbol{d}\rangle|. \tag{54}$$

Define events

$$\mathcal{E}_1 = \left\{ \left| \sum_{i=1}^n (y_i - \alpha x_i) e_i^{(\mathbf{c})} \right| \leq t_1 : \ \mathbf{c} = 1, \dots, 2^R \right\}, \tag{55}$$

and

$$\mathcal{E}_2 = \left\{ \left| \sum_{i=1}^n (y_i - \alpha x_i) e_i^{(\tilde{\boldsymbol{x}})} \right| \leq t_2 \right\}. \tag{56}$$

Conditioned on $\mathcal{E}_1 \cap \mathcal{E}_2$, it follows that

$$\|\mathbf{e}\|^2 \leq \|\boldsymbol{d}\|^2 + 2(t_1 + t_2)/\alpha, \tag{57}$$

Using Lemma 2 with $y_i \sim \mathrm{Poisson}(\alpha x_i)$ and $w_i = e_i^{(\mathbf{c})}$, it follows that

$$\mathbb{P}\left( \left| \sum_{i=1}^n (y_i - \alpha x_i) e_i^{(\mathbf{c})} \right| \geq t \right) \leq 2 \exp\left( -\frac{t^2}{2(\sigma_n^2 + w_M t/3)} \right), \tag{58}$$

where

$$\sigma_n^2 = \sum_{i=1}^n w_i^2 \alpha_i = \alpha \sum_{i=1}^n (e_i^{(\mathbf{c})})^2 x_i \leq n \alpha x_{\max}^3, \tag{59}$$

and

$$w_M = \max_{i \in \{1, \dots, n\}} |w_i| \leq x_{\max}.$$

Using the union bound and noting that $|\mathcal{C}| \leq 2^R$, we have

$$\mathbb{P}(\mathcal{E}_1^c) \leq 2^{R+1} \exp\left( -\frac{t_1^2}{2(n x_{\max}^3 \alpha + t_1 x_{\max}/3)} \right) \tag{60}$$

and

$$\mathbb{P}(\mathcal{E}_2^c) \leq 2 \exp\left( -\frac{t_2^2}{2(n x_{\max}^3 \alpha + t_2 x_{\max}/3)} \right). \tag{61}$$

Setting $t_1$ and $t_2$ such that they are both smaller than $3 n x_{\max}^2 \alpha$, and noting that $x_{\max} < 1$, we have

$$\mathbb{P}(\mathcal{E}_1^c \cap \mathcal{E}_2^c) \leq 2^{R+1} \exp\left(-\frac{t_1^2}{4n\alpha}\right) + 2 \exp\left(-\frac{t_2^2}{4n\alpha}\right). \tag{62}$$

For $\eta \in (0, 1)$, set

$$t_1 = 2\sqrt{n(\ln 2)(1+\eta)\alpha R},$$

and

$$t_2 = 2\sqrt{n\eta(\ln 2)\alpha R}.$$

Then,

$$\mathbb{P}(\mathcal{E}_1^c \cup \mathcal{E}_2^c) \leq 2^{-\eta R + 2}. \tag{63}$$

Using the selected values of $t_1$ and $t_2$ in (57) yields the desired result, i.e.,

$$\|\mathbf{e}\|_2^2 \leq \|\boldsymbol{d}\|_2^2 + 4\sqrt{n(\ln 2)(1+\eta)\frac{R}{\alpha}} + 4\sqrt{n\eta(\ln 2)\frac{R}{\alpha}}$$

$$\leq \|\boldsymbol{d}\|_2^2 + \sqrt{\frac{nR}{\alpha}}\left(4\sqrt{\ln 2}\right)\left(\sqrt{1+\eta} + \sqrt{\eta} + 1\right).$$

$\qquad\qquad\qquad\qquad\qquad\qquad\qquad\qquad\qquad\qquad\qquad\qquad\qquad\qquad\qquad\qquad\quad \square$

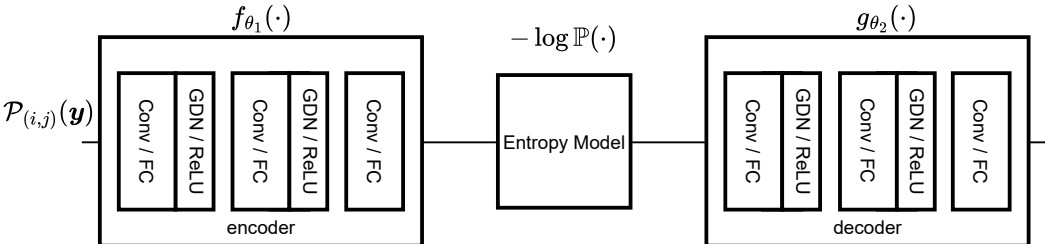

Figure 5: Neural compression network used for denoising. Conv and FC denote the convolutiona and fully connected layer, respectively. GDN and ReLU are activation functions.

# B  Additional experiments and experimental settings

In this section, we provide the details of the networks structures and experimental settings. We also present more experiments for Poisson denoising using MLE and MSE loss functions with unknown noise level.

## B.1  Network structure

For our experiments we used 3 convolutional layers in the encoder with 128 number of channels for the first two layers in the encoder (and the last two layers of decoder), see Figure 5. For color images we choose the number of channels in the last encoder (and first decoder) layer equals to 32, and for grayscale images equals to 16. The MLP-based network of the ablation study in Section 5 has 3 fully connected layers in the encoder with 1024 hidden units for the first two layers in the encoder (and the last two layers of decoder). The number of hidden units in the last encoder (and first decoder) layer equals to 16. As activation function we use GDN [32] for Conv network and ReLU for MLP.

## B.2  More recent improvements on Zero-shot Noise2Noise

Zero-shot Noise2Noise (ZS-N2N) [24] learns a mapping between two downsampled noisy images extracted from the original image. This strategy of generating noisy samples was extended in two recent concurrent works. Dual-sampling Noise2Noise (DS-N2N) [29] identifies a key sub-optimality in ZS-N2N: the network learns to denoise at a lower resolution but is then directly applied to the larger, original resolution. To address this, the authors proposed to apply an additional bicubic upsampling to the downsampled images and train the network on both the low-resolution and the upsampled pairs, which boosts the performance of ZS-N2N.

In Pixel2Pixel [30], the authors identified the local sampling in ZS-N2N insufficient to break the spatial correlation of *real-world noise*. To address this issue, Pixel2Pixel first finds a "pixel bank" for each pixel based on the non-local similar patches. Then, by randomly choosing pixels from the bank, it generates multiple noisy pairs to be used as a pseudo-training dataset for zero-shot denoising.

We have compared the performance of both methods under AWGN and Poisson noise in Tables 5 and 6, respectively. Each paper's official code was used to report the numbers in the tables. In AWGN, our compression-based denoiser ZS-NCD outperforms both methods while achieving the second-best performance in Poisson denoising. Pixel2Pixel performs well in Poisson denoising.

Table 5: Average PSNR(dB)/SSIM denoising performance on Kodak24 dataset under AWGN $\mathcal{N}(0, \sigma_z^2)$. Best results are in **bold**, second-best are underlined.

|  | $\sigma_z = 15$ | $\sigma_z = 25$ | $\sigma_z = 50$ |
|---|---|---|---|
| ZS-N2N (2023) | 32.30 / 0.8650 | 29.54 / 0.7798 | 25.82 / 0.6151 |
| DS-N2N (2025) | 32.31 / 0.8803 | 29.64 / 0.8044 | 25.42 / 0.6378 |
| Pixel2Pixel (2025) | 31.31 / 0.8707 | 29.89 / 0.8098 | 26.55 / 0.6873 |
| **ZS-NCD** | **33.18 / 0.9026** | **30.60 / 0.8144** | **27.89 / 0.7464** |

Table 6: Average PSNR(dB)/SSIM denoising performance on Kodak24 dataset under Poison noise, Poisson($\alpha x$)/$\alpha$. Best results are in **bold**, second-best are underlined.

|  | $\alpha = 15$ | $\alpha = 25$ | $\alpha = 50$ |
|---|---|---|---|
| ZS-N2N (2023) | 26.80 / 0.6757 | 28.21 / 0.7374 | 30.13 / 0.8076 |
| DS-N2N (2025) | 27.29 / 0.7016 | 28.50 / 0.7540 | 30.27 / 0.8250 |
| Pixel2Pixel (2025) | **28.22 / 0.7390** | **29.26 / 0.7891** | 30.41 / **0.8372** |
| **ZS-NCD** | 27.64 / 0.7432 | 28.77 / 0.7677 | **30.60** / 0.8235 |

## B.3 MSE and likelihood estimation under Poisson noise without knowing true $\alpha$

We compare the MSE and MLE distortion for Poisson denoising in Table 7 using the estimated $\hat{\alpha}$ as explained in Section 4.

Table 7: Minimizing Poisson negative log-likelihood (NLL) vs. MSE with estimated $\hat{\alpha}$ for *Cameraman* image in Set11. PSNR / SSIM are reported here.

| $\alpha$ | MSE (with estimated $\hat{\alpha}$) | NLL (with estimated $\hat{\alpha}$) |
|---|---|---|
| 15 | 23.41 / 0.7554 | 23.13 / 0.7567 |
| 50 | 25.22 / 0.7961 | 24.88 / 0.7460 |

## B.4 Study on factors in patch-wise compression affecting denoising

In this section, we explain the intuition behind why learning compression networks and denoising on overlapped patches is feasible. The centered pixels in the patches are better compressed as empirically observed in [58], thus they can provide better denoising performance. To study the contribution of each patch containing the single pixel to be denoised we design the experiment that, in the denoising phase, we denoise the overlapped patches, but only a single pixel at the same location from each patch is used to construct the final denoised image, instead of averaging all of them as in (9). We show the denoising performance of each pixel location in Figure 6. The PSNR at each pixel denotes the denoising performance of only using the specific pixel of each overlapped patches with stride 1. We can find that the boundary pixels give lower PSNR, which is consistent with previous research findings that the centered pixels are better compressed. Next, we analyze the effect of patch size in both learning and denoising phases. Given that scaler quantization is applied and the entropy model is learned on latent code of the patches, the compression performance on the latent code is affected by both the patch size and the number of downsampling operations in CNN-based encoder. We design the experiment that 3 downsampling operators are applied to patch size 8 and 16, where the latent code sizes are $1 \times 1 \times n_b$ and $2 \times 2 \times n_b$ respectively, where the denoising performance at each pixel location is in Figure 6 (**Left**) and (**Middle**), and if we increase the downsampling to 4 for patch size 16, which results in the latent code size to be $1 \times 1 \times n_b$, the denoising performance is in Figure 6 (**Right**). We find that spatial size of the latent code to be quantized matters given the scaler quantization limitation, the reconstructed output by the decoder will be restricted by the only correlated latent code as we can observe. Motivated from this, we perform the learning and denoising phases both patch-wise with proper networks structure, all pixels in each patch are used and the overlapped areas are averaged properly to reduce the variance of the compression-based estimates.

## C Additional numerical results

In this section, we provide the full denoising numerical results of the denoisers on all the test images. All the experiments were run on Nvidia RTX 6000 Ada with 48 GB memory. It takes 40 minutes to denoise a grayscale image of size $256 \times 256$, and 50 minutes for an RGB image of size $512 \times 768$. Adam optimizer is used for training the networks over 20K steps, with initial learning rate of $5 \times 10^{-3}$ decreased to $5 \times 10^{-4}$ after 16K steps for the Conv-based network. The learning rate for MLP-based networks is $1 \times 10^{-3}$.

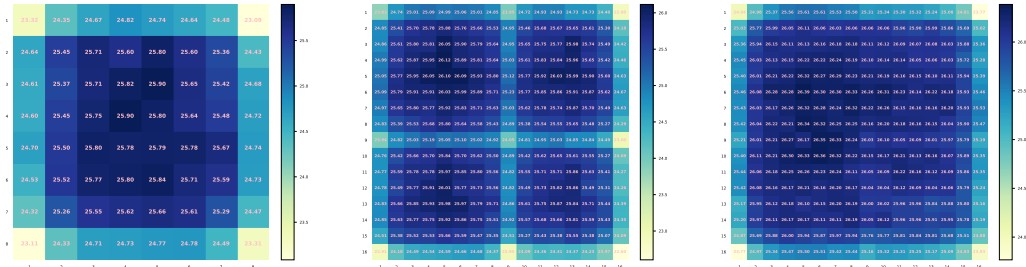

Figure 6: Denoising AWGN ($\sigma_z = 25$) of image *Parrot* by only compressing a single pixel in each overlapped patches with stride 1. The PSNR at each pixel denotes the final denoising performance by only compressing the pixel at that specific location in each patch. **Left**: patch size $8 \times 8$, with downsampling factor equals 8 in $f_{\theta_1}$; **Middle**: patch size $16 \times 16$, with downsampling factor equals 8 in $f_{\theta_1}$; **Right**: patch size $16 \times 16$, with downsampling factor equals 16 in $f_{\theta_1}$.

## C.1 Set11 Dataset

For noise levels $(15, 25, 50)$ we set $\lambda = (300, 850, 3000)$. Similar to Kodak and other experiments we set training epochs to have 20K steps of gradient back propagation. For Poisson denoising $\alpha = (15, 25, 50)$ the $\lambda = (3000, 1500, 1000)$. We report the detailed results of AWGN denoising in Table 8, and Poisson noise denoising in Table 9.

Table 8: Set11 Denoising performance comparison under AWGN $\mathcal{N}(0, \sigma_z^2 I)$.

| | | 256 × 256 | | | | | | | 512 × 512 | | | | |
|---|---|---|---|---|---|---|---|---|---|---|---|---|---|
| $\sigma$ | Method | C.man | House | Peppers | Starfish | Monarch | Airplane | Parrot | Barbara | Boats | Pirate | Couple | Average |
| | BM3D | 31.84/0.8974 | 34.94/0.8870 | 32.82/0.9118 | 31.16/0.9082 | 31.92/0.9409 | 31.09/0.9034 | 31.47/0.9032 | 33.04/0.9253 | 32.12/0.8604 | 31.94/0.8726 | 32.11/0.8795 | 32.22/0.8991 |
| | JPEG2K | 27.12/0.7474 | 29.48/0.7621 | 27.96/0.7907 | 26.75/0.8077 | 26.74/0.8166 | 26.58/0.7664 | 27.30/0.7778 | 26.76/0.7690 | 27.87/0.7390 | 27.92/0.7471 | 27.44/0.7449 | 27.45/0.7699 |
| | DIP | 27.94/0.7417 | 31.39/0.8111 | 29.80/0.8273 | 29.58/0.8605 | 29.93/0.8767 | 28.14/0.8047 | 28.37/0.7794 | 27.65/0.7538 | 29.48/0.7798 | 29.27/0.7817 | 28.65/0.7727 | 29.11/0.7990 |
| | DD | 29.41/0.8099 | 32.83/0.8406 | 26.97/0.8488 | 29.39/0.8739 | 30.01/0.8957 | 26.44/0.8228 | 29.32/0.8447 | 24.48/0.7089 | 29.45/0.7883 | 29.78/0.8085 | 29.06/0.7938 | 28.83/0.8215 |
| 15 | ZS-N2N | 30.14/0.8133 | 32.19/0.8158 | 30.58/0.8264 | 29.52/0.8639 | 30.15/0.8551 | 29.98/0.8298 | 30.19/0.8290 | 27.70/0.7772 | 30.06/0.7900 | 30.06/0.7957 | 29.59/0.7913 | 30.01/0.8169 |
| | ZS-N2S | 27.66/0.8272 | 31.08/0.8442 | 29.46/0.8675 | 28.83/0.8810 | 28.77/0.8961 | 27.34/0.8591 | 27.67/0.8528 | 28.75/0.8534 | 29.52/0.8139 | 29.41/0.8181 | 29.60/0.8311 | 28.92/0.8495 |
| | S2S | 20.29/0.6769 | 32.96/0.8633 | 23.96/0.8387 | 25.50/0.8250 | 30.05/0.9269 | 28.10/0.8611 | 20.20/0.7132 | 30.35/0.8865 | 27.74/0.7871 | 29.97/0.8192 | 25.82/0.7754 | 26.81/0.8158 |
| | **ZS-NCD** | 30.83/0.8554 | 34.45/0.8835 | 32.20/0.8844 | 31.34/0.8749 | 31.83/0.8966 | 30.07/0.8552 | 30.40/0.8464 | 31.14/0.8826 | 31.09/0.8014 | 30.82/0.8302 | 30.67/0.8279 | 31.35/0.8580 |
| | **ZS-NCD (MLP)** | 31.18/0.8680 | 34.86/0.8887 | 32.43/0.9009 | 31.37/0.9053 | 32.04/0.9263 | 30.79/0.8848 | 31.02/0.8839 | 32.55/0.9123 | 31.82/0.8468 | 31.35/0.8552 | 31.55/0.8638 | 31.91/0.8851 |
| | BM3D | 29.54/0.8499 | 32.79/0.8561 | 30.13/0.8705 | 28.58/0.8578 | 29.36/0.9046 | 28.50/0.8584 | 28.94/0.8561 | 30.63/0.8887 | 29.87/0.8039 | 29.64/0.8082 | 29.74/0.8206 | 29.79/0.8523 |
| | JPEG2K | 24.49/0.6976 | 27.26/0.7269 | 24.93/0.7206 | 24.18/0.7167 | 24.06/0.7561 | 23.91/0.7126 | 24.38/0.7162 | 24.09/0.6825 | 25.61/0.6577 | 25.76/0.6566 | 25.28/0.6534 | 24.91/0.6997 |
| | DIP | 25.23/0.6043 | 28.93/0.7545 | 27.39/0.7579 | 26.39/0.7777 | 27.47/0.8169 | 25.57/0.6983 | 26.29/0.7409 | 24.75/0.6356 | 27.05/0.6843 | 27.06/0.6857 | 26.52/0.6847 | 26.60/0.7128 |
| | DD | 27.24/0.7521 | 30.48/0.8023 | 25.39/0.7591 | 26.86/0.8051 | 27.69/0.8526 | 24.93/0.7120 | 27.29/0.7863 | 23.81/0.6455 | 27.49/0.7163 | 27.87/0.7380 | 27.13/0.7131 | 26.93/0.7530 |
| 25 | ZS-N2N | 27.32/0.7089 | 29.36/0.7276 | 27.46/0.7240 | 26.61/0.7821 | 27.20/0.7634 | 27.02/0.7463 | 27.16/0.7149 | 25.49/0.6854 | 27.26/0.6779 | 27.48/0.6931 | 26.63/0.6673 | 27.18/0.7173 |
| | ZS-N2S | 26.24/0.7843 | 29.23/0.8073 | 27.77/0.8233 | 27.61/0.8463 | 27.35/0.8569 | 25.86/0.8023 | 26.27/0.7997 | 26.43/0.7759 | 28.23/0.7580 | 27.52/0.7526 | 27.74/0.7617 | 27.30/0.7971 |
| | S2S | 16.93/0.5998 | 29.12/0.8275 | 21.88/0.7666 | 21.14/0.6974 | 25.93/0.8606 | 24.12/0.7350 | 17.09/0.6069 | 25.79/0.7980 | 23.94/0.7061 | 27.32/0.7403 | 23.29/0.6979 | 23.32/0.7306 |
| | **ZS-NCD** | 28.78/0.8237 | 32.14/0.8547 | 29.62/0.8406 | 28.48/0.8134 | 29.02/0.8494 | 27.77/0.8126 | 28.14/0.8007 | 28.39/0.8192 | 28.85/0.7444 | 28.64/0.7630 | 28.37/0.7648 | 28.93/0.8079 |
| | **ZS-NCD (MLP)** | 29.08/0.8259 | 32.63/0.8525 | 29.85/0.8574 | 28.73/0.8547 | 29.42/0.8861 | 28.37/0.8477 | 28.75/0.8431 | 30.01/0.8658 | 29.59/0.7843 | 29.15/0.7859 | 29.10/0.7958 | 29.52/0.8363 |
| | BM3D | 26.56/0.7813 | 29.61/0.8029 | 26.85/0.7911 | 25.07/0.7508 | 25.82/0.8192 | 25.29/0.7713 | 26.02/0.7809 | 27.02/0.7888 | 26.76/0.7003 | 26.75/0.6962 | 26.37/0.6977 | 26.56/0.7619 |
| | JPEG2K | 21.49/0.5880 | 24.24/0.6444 | 21.72/0.6077 | 21.39/0.5784 | 20.86/0.6414 | 21.11/0.6021 | 21.29/0.6035 | 21.65/0.5377 | 22.83/0.5318 | 23.29/0.5356 | 22.67/0.5025 | 22.05/0.5794 |
| | DIP | 22.73/0.5846 | 25.67/0.6475 | 23.81/0.5987 | 22.99/0.6406 | 23.06/0.6293 | 22.64/0.5522 | 23.02/0.5811 | 22.38/0.5316 | 23.90/0.5371 | 24.43/0.5524 | 23.40/0.5064 | 23.46/0.5783 |
| | DD | 23.89/0.6487 | 27.27/0.7282 | 22.95/0.7276 | 23.44/0.6700 | 23.55/0.7319 | 22.52/0.6652 | 23.87/0.6471 | 22.72/0.5980 | 24.47/0.6050 | 25.19/0.6340 | 24.30/0.5872 | 24.01/0.6584 |
| 50 | ZS-N2N | 23.36/0.5324 | 25.17/0.5167 | 23.86/0.5669 | 22.92/0.6186 | 22.95/0.6010 | 23.39/0.5988 | 22.87/0.5136 | 22.62/0.5150 | 23.93/0.5138 | 24.30/0.5330 | 23.30/0.4930 | 23.52/0.5457 |
| | ZS-N2S | 24.65/0.6966 | 26.72/0.7091 | 25.24/0.7297 | 24.05/0.7102 | 24.82/0.7618 | 24.04/0.7467 | 24.00/0.7078 | 22.81/0.5916 | 25.46/0.6512 | 25.55/0.6469 | 24.80/0.6197 | 24.74/0.6883 |
| | S2S | 14.23/0.4809 | 21.14/0.6396 | 17.80/0.5763 | 15.71/0.4176 | 18.33/0.5955 | 15.70/0.4828 | 13.66/0.4446 | 17.60/0.4883 | 18.69/0.5264 | 19.55/0.5354 | 19.12/0.5325 | 17.41/0.5200 |
| | **ZS-NCD** | 25.55/0.7616 | 28.62/0.7995 | 26.31/0.7604 | 24.59/0.6925 | 25.53/0.7585 | 24.65/0.7338 | 25.31/0.7228 | 24.06/0.6525 | 25.61/0.6538 | 25.92/0.6707 | 25.19/0.6519 | 25.58/0.7144 |
| | **ZS-NCD (MLP)** | 25.61/0.7342 | 29.13/0.7881 | 26.30/0.7702 | 24.85/0.7426 | 25.12/0.7795 | 24.84/0.7497 | 25.24/0.7596 | 25.55/0.7146 | 26.26/0.6698 | 26.25/0.6717 | 25.65/0.6560 | 25.89/0.7306 |

Table 9: Set11 Denoising performance comparison under Poisson noise $\mathrm{Poisson}(\alpha \boldsymbol{x})/\alpha$.

| | | 256 × 256 | | | | | | | 512 × 512 | | | | |
|---|---|---|---|---|---|---|---|---|---|---|---|---|---|
| $\alpha$ | Method | C.man | House | Peppers | Starfish | Monarch | Airplane | Parrot | Barbara | Boats | Pirate | Couple | Average |
| | BM3D | 26.64/0.7651 | 29.39/0.7668 | 27.13/0.7914 | 24.93/0.7519 | 26.32/0.8265 | 24.79/0.6730 | 26.26/0.7866 | 27.24/0.7860 | 26.82/0.6977 | 27.07/0.7048 | 26.67/0.7056 | 26.66/0.7505 |
| | JPEG2K | 21.98/0.6032 | 24.35/0.6106 | 22.12/0.6213 | 21.52/0.5887 | 21.24/0.6493 | 20.87/0.5818 | 22.02/0.6378 | 21.94/0.5688 | 23.09/0.5346 | 23.75/0.5510 | 23.01/0.5237 | 22.35/0.5882 |
| | DIP | 22.85/0.5382 | 26.32/0.6528 | 24.23/0.6138 | 23.23/0.6696 | 23.54/0.6875 | 22.07/0.4938 | 24.38/0.6830 | 22.89/0.6081 | 24.64/0.6023 | 25.61/0.6516 | 24.58/0.5986 | 24.37/0.6629 |
| 15 | ZS-N2N | 24.19/0.5818 | 25.41/0.5346 | 24.65/0.6016 | 23.12/0.6520 | 23.92/0.6441 | 23.12/0.5565 | 23.83/0.5821 | 23.05/0.5503 | 24.40/0.5403 | 24.87/0.5684 | 23.94/0.5305 | 24.04/0.5766 |
| | ZS-N2S | 24.94/0.7241 | 27.29/0.7317 | 25.71/0.7431 | 24.41/0.7417 | 25.37/0.7968 | 23.05/0.7051 | 24.98/0.7315 | 22.87/0.6087 | 26.08/0.6696 | 25.94/0.6655 | 25.09/0.6389 | 25.06/0.7051 |
| | S2S | 23.53/0.7325 | 22.01/0.7409 | 22.73/0.7300 | 18.20/0.6010 | 21.81/0.7813 | 16.18/0.5010 | 20.27/0.7304 | 22.10/0.7261 | 23.49/0.6529 | 24.72/0.6782 | 24.17/0.6843 | 21.75/0.6872 |
| | **ZS-NCD** | 25.73/0.7660 | 28.87/0.8015 | 26.54/0.7745 | 24.65/0.6988 | 25.86/0.7791 | 24.21/0.6568 | 25.52/0.7356 | 24.11/0.6562 | 25.48/0.6510 | 25.93/0.6705 | 25.29/0.6552 | 25.65/0.7132 |
| | BM3D | 22.69/0.5154 | 22.82/0.4765 | 22.74/0.5930 | 22.11/0.6947 | 23.44/0.7213 | 19.60/0.3788 | 23.05/0.5991 | 23.09/0.6508 | 22.77/0.5123 | 23.89/0.5979 | 23.45/0.5753 | 22.70/0.5741 |
| | JPEG2K | 22.54/0.6267 | 24.97/0.6773 | 22.87/0.6076 | 22.26/0.6378 | 22.55/0.6641 | 21.59/0.5649 | 22.62/0.6373 | 22.55/0.5976 | 23.71/0.5685 | 24.20/0.5801 | 23.49/0.5566 | 23.03/0.6108 |
| | DIP | 24.21/0.5976 | 27.06/0.6553 | 25.76/0.6945 | 24.41/0.7312 | 25.21/0.7384 | 23.58/0.6290 | 24.69/0.6608 | 23.11/0.5903 | 25.24/0.6130 | 26.10/0.6528 | 24.91/0.5998 | 24.94/0.6512 |
| 25 | ZS-N2N | 25.59/0.6334 | 27.14/0.6234 | 25.82/0.6522 | 24.33/0.7158 | 25.51/0.7109 | 24.53/0.6274 | 25.55/0.6617 | 24.09/0.6173 | 25.60/0.6018 | 26.11/0.6354 | 25.16/0.5958 | 25.40/0.6432 |
| | ZS-N2S | 26.22/0.7776 | 27.81/0.7643 | 26.55/0.7768 | 24.82/0.7795 | 26.48/0.8254 | 24.77/0.7463 | 25.44/0.7839 | 23.24/0.6387 | 27.25/0.7143 | 27.13/0.7200 | 26.44/0.6986 | 26.01/0.7478 |
| | S2S | 25.09/0.7572 | 24.10/0.7398 | 24.91/0.7733 | 19.11/0.6491 | 23.64/0.8226 | 17.93/0.6279 | 21.13/0.7692 | 24.01/0.7860 | 25.30/0.7058 | 26.45/0.7232 | 25.77/0.7360 | 23.40/0.7355 |
| | **ZS-NCD** | 27.17/0.7635 | 30.09/0.8109 | 27.92/0.7932 | 26.27/0.7600 | 27.28/0.8093 | 24.93/0.6116 | 26.74/0.7551 | 26.24/0.7393 | 27.30/0.6993 | 27.32/0.7192 | 26.83/0.7123 | 27.10/0.7431 |
| | BM3D | 22.94/0.5314 | 22.89/0.4548 | 23.22/0.5844 | 23.06/0.7150 | 23.87/0.6990 | 20.51/0.4115 | 23.65/0.6136 | 23.54/0.6545 | 22.82/0.5205 | 23.97/0.6087 | 23.47/0.5723 | 23.09/0.5787 |
| | JPEG2K | 24.23/0.6635 | 26.87/0.6796 | 24.96/0.7042 | 24.08/0.7240 | 24.05/0.7568 | 23.40/0.6387 | 24.57/0.7077 | 23.95/0.6804 | 25.37/0.6414 | 25.73/0.6483 | 25.25/0.6475 | 24.77/0.6811 |
| | DIP | 25.34/0.6348 | 28.88/0.7369 | 27.59/0.7559 | 26.18/0.7891 | 26.58/0.7728 | 24.69/0.6457 | 26.04/0.7062 | 23.88/0.6158 | 26.61/0.6712 | 27.21/0.7015 | 26.34/0.6747 | 26.30/0.7004 |
| 50 | ZS-N2N | 27.24/0.7398 | 30.16/0.7784 | 25.44/0.7615 | 26.78/0.8127 | 27.82/0.8527 | 24.39/0.6568 | 27.39/0.7735 | 23.86/0.6518 | 27.30/0.7096 | 28.03/0.7476 | 27.16/0.7162 | 26.87/0.7455 |
| | ZS-N2S | 27.63/0.7210 | 29.30/0.7113 | 27.92/0.7424 | 26.35/0.7857 | 27.38/0.7723 | 26.26/0.7037 | 27.68/0.7443 | 25.50/0.6883 | 27.31/0.6832 | 27.68/0.7075 | 26.84/0.6783 | 27.26/0.7216 |
| | S2S | 26.82/0.8041 | 29.32/0.7832 | 27.75/0.8192 | 26.62/0.8243 | 28.03/0.8624 | 26.05/0.8097 | 26.56/0.8196 | 23.60/0.6606 | 27.92/0.7493 | 27.68/0.7557 | 27.54/0.7521 | 27.08/0.7855 |
| | S2S | 26.72/0.8220 | 27.19/0.8106 | 27.83/0.8300 | 20.47/0.7126 | 26.38/0.8780 | 21.09/0.6607 | 22.61/0.8001 | 27.07/0.8457 | 27.39/0.7620 | 28.31/0.7754 | 27.64/0.7885 | 25.70/0.7896 |
| | **ZS-NCD** | 28.24/0.8093 | 31.90/0.8488 | 29.44/0.8410 | 28.02/0.8064 | 28.78/0.8504 | 26.99/0.7422 | 27.93/0.7961 | 27.60/0.7920 | 28.16/0.7273 | 28.11/0.7466 | 27.72/0.7456 | 28.44/0.7914 |

## C.2 Set13 Dataset

All images are center-cropped at size of $192 \times 192$. For this set of images we set $\lambda = (100, 200, 800)$ and for noise levels $\sigma_z = (15, 25, 50)$ and for Poisson denoising we have $\lambda = (900, 500, 200)$ for noise levels $\alpha = (15, 25, 50)$. We report the detailed results of AWGN denoising in Table 10, and Poisson noise denoising in Table 11.

Table 10: Set13 Denoising performance comparison under AWGN $\mathcal{N}(0, \sigma_z^2 I)$.

| $\sigma$ | Method | Baboon | Barbara | Bridge | Coastguard | Comic | Face | Flowers | Foreman | Man | Monarch | Peppers | PPT3 | Zebra | Average |
|---|---|---|---|---|---|---|---|---|---|---|---|---|---|---|---|
| 15 | BM3D | 28.56/0.7797 | 33.07/0.9151 | 30.39/0.8723 | 30.18/0.8799 | 28.74/0.9289 | 30.28/0.7665 | 29.62/0.9040 | 29.88/0.8323 | 31.13/0.9361 | 31.77/0.8199 | 34.49/0.9588 | 31.02/0.9198 | 31.15/0.8808 | |
| | JPEG2K | 25.12/0.6539 | 27.01/0.7674 | 26.27/0.7426 | 25.79/0.7323 | 24.86/0.8017 | 27.35/0.6482 | 25.52/0.7986 | 30.86/0.8397 | 25.97/0.6983 | 25.63/0.7649 | 28.27/0.7161 | 28.00/0.8183 | 26.33/0.8243 | 26.69/0.7543 |
| | DIP | 27.25/0.7498 | 30.92/0.8403 | 30.18/0.8692 | 30.79/0.9036 | 28.25/0.9091 | 29.62/0.7611 | 28.85/0.8817 | 33.81/0.8947 | 29.99/0.8333 | 31.18/0.9085 | 30.30/0.7782 | 32.22/0.8908 | 30.63/0.9203 | 30.31/0.8570 |
| | DD | 26.35/0.7029 | 24.27/0.7066 | 29.16/0.8508 | 28.80/0.8421 | 26.44/0.8932 | 29.59/0.7398 | 27.34/0.8634 | 34.87/0.9274 | 28.68/0.8073 | 30.02/0.9151 | 31.07/0.8003 | 33.09/0.9277 | 30.22/0.9052 | 29.22/0.8371 |
| | ZS-N2N | 28.63/0.7992 | 28.45/0.7803 | 32.08/0.9041 | 31.54/0.9141 | 28.70/0.9018 | 30.64/0.8048 | 29.67/0.8955 | 34.02/0.8817 | 31.63/0.8801 | 31.65/0.9120 | 31.28/0.8089 | 32.84/0.9009 | 31.27/0.9277 | 30.95/0.8701 |
| | ZS-N2S | 20.92/0.5844 | 21.14/0.5730 | 21.37/0.6468 | 20.78/0.4893 | 15.80/0.4987 | 22.03/0.5641 | 17.49/0.5154 | 8.43/0.3637 | 22.69/0.6725 | 12.21/0.6124 | 21.60/0.6438 | 11.58/0.4462 | 20.37/0.7871 | 18.18/0.5690 |
| | S2S | 22.36/0.5810 | 30.39/0.8769 | 22.74/0.7485 | 22.72/0.7108 | 17.44/0.7015 | 17.23/0.4383 | 21.14/0.7121 | 16.78/0.8102 | 15.65/0.4463 | 26.92/0.8838 | 24.20/0.7398 | 15.48/0.7528 | 14.84/0.5400 | 20.61/0.6879 |
| | **ZS-NCD** | 28.10/0.7831 | 33.85/0.9208 | 31.49/0.9051 | 32.65/0.9345 | 29.23/0.9355 | 30.51/0.7891 | 29.69/0.9077 | 35.85/0.9381 | 31.49/0.8867 | 33.21/0.9445 | 31.60/0.8281 | 35.07/0.9601 | 32.37/0.9448 | 31.93/0.8983 |
| 25 | BM3D | 26.56/0.6892 | 30.70/0.8824 | 28.06/0.7955 | 27.59/0.7664 | 25.90/0.8698 | 28.87/0.6978 | 26.93/0.8396 | 33.51/0.9087 | 27.35/0.7428 | 28.72/0.9056 | 30.11/0.7823 | 31.66/0.9324 | 28.61/0.8643 | 28.81/0.8213 |
| | JPEG2K | 23.98/0.5711 | 24.14/0.6801 | 24.14/0.6400 | 23.53/0.5801 | 21.36/0.7014 | 26.17/0.5692 | 22.77/0.6809 | 28.30/0.7960 | 23.51/0.5667 | 23.03/0.7101 | 26.47/0.6729 | 25.23/0.7707 | 23.58/0.7389 | 24.32/0.6676 |
| | DIP | 25.70/0.6734 | 27.27/0.7021 | 27.79/0.7999 | 27.86/0.8126 | 25.11/0.8382 | 28.16/0.6599 | 26.03/0.8164 | 31.17/0.8384 | 27.27/0.7446 | 29.02/0.8785 | 28.82/0.7313 | 29.20/0.8106 | 28.65/0.8820 | 27.85/0.7837 |
| | DD | 25.56/0.6589 | 23.56/0.6676 | 26.94/0.7758 | 26.77/0.7542 | 24.76/0.8397 | 28.37/0.6819 | 25.72/0.8058 | 32.22/0.8849 | 27.09/0.7366 | 27.68/0.8566 | 29.42/0.7615 | 29.94/0.8937 | 28.11/0.8641 | 27.40/0.7832 |
| | ZS-N2N | 26.85/0.7287 | 26.73/0.7136 | 29.00/0.8282 | 28.30/0.8312 | 25.95/0.8462 | 28.71/0.7250 | 26.67/0.8248 | 31.31/0.8156 | 28.88/0.7940 | 29.03/0.8683 | 28.95/0.7282 | 29.78/0.8205 | 28.54/0.8768 | 28.36/0.8001 |
| | ZS-N2S | 19.22/0.4963 | 15.44/0.3973 | 22.05/0.5859 | 21.55/0.5581 | 16.64/0.5734 | 25.76/0.6423 | 17.98/0.6324 | 20.90/0.6792 | 22.30/0.6170 | 21.29/0.7778 | 22.45/0.6535 | 17.04/0.6502 | 22.41/0.7964 | 20.39/0.6200 |
| | S2S | 18.66/0.4947 | 24.97/0.7969 | 19.43/0.6054 | 20.69/0.5890 | 16.30/0.6168 | 15.08/0.3846 | 17.57/0.5267 | 15.53/0.7605 | 14.11/0.3892 | 21.34/0.7849 | 22.40/0.6795 | 13.60/0.6805 | 13.69/0.4890 | 17.95/0.5998 |
| | **ZS-NCD** | 26.54/0.7128 | 30.65/0.8388 | 28.78/0.8332 | 29.46/0.8718 | 26.93/0.8847 | 29.18/0.7367 | 27.43/0.7956 | 34.55/0.9108 | 28.83/0.8027 | 30.18/0.8917 | 29.62/0.7656 | 31.56/0.8898 | 29.63/0.9023 | 29.33/0.8351 |
| 50 | BM3D | 24.66/0.5953 | 26.99/0.7854 | 25.31/0.6606 | 24.45/0.5245 | 22.30/0.7299 | 27.19/0.6140 | 23.52/0.7068 | 29.84/0.8400 | 24.77/0.6272 | 25.62/0.8492 | 27.34/0.7139 | 27.51/0.8657 | 25.59/0.7613 | 25.78/0.7134 |
| | JPEG2K | 21.99/0.4807 | 21.81/0.5541 | 21.55/0.4898 | 21.41/0.3641 | 18.50/0.5254 | 23.43/0.4603 | 19.92/0.5254 | 24.51/0.6873 | 20.74/0.4169 | 19.95/0.5819 | 23.60/0.5810 | 21.26/0.6490 | 19.81/0.5670 | 21.43/0.5295 |
| | DIP | 24.07/0.5778 | 23.06/0.5943 | 25.05/0.6773 | 24.10/0.6179 | 21.64/0.7142 | 26.07/0.5804 | 22.84/0.6871 | 28.34/0.7754 | 24.60/0.6173 | 25.68/0.7922 | 26.06/0.6398 | 25.60/0.7066 | 25.55/0.7926 | 24.82/0.6748 |
| | DD | 23.87/0.5745 | 23.06/0.6436 | 24.01/0.6402 | 23.43/0.4732 | 21.37/0.6956 | 26.83/0.6127 | 22.73/0.6819 | 29.12/0.8434 | 24.08/0.6011 | 24.09/0.7616 | 26.82/0.7055 | 25.40/0.8137 | 24.41/0.7659 | 24.56/0.6779 |
| | ZS-N2N | 24.41/0.5999 | 23.99/0.5573 | 25.19/0.6726 | 24.83/0.6513 | 21.87/0.7014 | 25.90/0.5732 | 22.72/0.6619 | 27.10/0.6419 | 24.91/0.6111 | 25.11/0.7571 | 25.35/0.5654 | 25.14/0.6243 | 24.62/0.7594 | 24.67/0.6444 |
| | ZS-N2S | 22.32/0.5704 | 16.65/0.5507 | 22.29/0.6475 | 21.75/0.4859 | 15.72/0.3989 | 25.49/0.6110 | 18.56/0.6121 | 24.16/0.7812 | 22.50/0.5681 | 19.83/0.6154 | 19.54/0.5440 | 16.26/0.4869 | 23.00/0.7724 | 20.62/0.5880 |
| | S2S | 14.08/0.3567 | 17.56/0.5046 | 14.95/0.3650 | 17.25/0.3204 | 13.55/0.3265 | 12.73/0.2916 | 13.31/0.2712 | 13.82/0.5715 | 12.16/0.2727 | 14.59/0.5444 | 17.16/0.4959 | 11.42/0.5186 | 12.16/0.2807 | 14.21/0.3938 |
| | **ZS-NCD** | 24.32/0.6035 | 26.84/0.7333 | 25.70/0.7172 | 25.76/0.7385 | 23.14/0.7768 | 26.90/0.6401 | 23.74/0.7212 | 28.46/0.7716 | 25.46/0.6710 | 26.29/0.8075 | 26.40/0.6722 | 27.25/0.7856 | 26.10/0.8113 | 25.87/0.7269 |

Table 11: Set13 Denoising performance comparison under Poisson noise $\mathrm{Poisson}(\alpha \boldsymbol{x})/\alpha$.

| $\alpha$ | Method | Baboon | Barbara | Bridge | Coastguard | Comic | Face | Flowers | Foreman | Man | Monarch | Peppers | PPT3 | Zebra | Average |
|---|---|---|---|---|---|---|---|---|---|---|---|---|---|---|---|
| 15 | BM3D | 24.46/0.5651 | 26.63/0.7712 | 25.18/0.6462 | 24.45/0.5186 | 22.07/0.7101 | 27.46/0.6342 | 23.51/0.7017 | 29.05/0.7747 | 24.98/0.6405 | 25.60/0.8101 | 27.53/0.7041 | 26.16/0.7288 | 26.19/0.7802 | 25.64/0.6912 |
| | JPEG2K | 22.01/0.4873 | 21.82/0.5515 | 21.77/0.5163 | 21.69/0.3727 | 18.47/0.5237 | 24.97/0.5289 | 20.42/0.5841 | 23.83/0.6996 | 21.69/0.4723 | 20.21/0.5809 | 24.17/0.6012 | 20.73/0.5858 | 21.04/0.6381 | 21.76/0.5494 |
| | DIP | 24.30/0.5907 | 23.05/0.6010 | 25.46/0.7069 | 24.21/0.5996 | 21.79/0.7282 | 26.99/0.6150 | 23.41/0.7456 | 28.03/0.7686 | 25.10/0.6444 | 26.34/0.8136 | 26.26/0.6510 | 25.44/0.7088 | 26.37/0.8180 | 25.14/0.6916 |
| | DD | 23.89/0.5750 | 23.16/0.6479 | 24.83/0.6873 | 23.80/0.5544 | 21.52/0.7107 | 27.22/0.6387 | 23.41/0.7339 | 29.25/0.8522 | 25.03/0.6514 | 24.67/0.7745 | 27.09/0.7156 | 25.12/0.7640 | 25.43/0.8020 | 24.96/0.7006 |
| | ZS-N2N | 24.82/0.6330 | 24.11/0.5768 | 25.82/0.7281 | 25.35/0.6946 | 21.90/0.7055 | 27.11/0.6779 | 23.67/0.7490 | 27.20/0.6491 | 26.60/0.7290 | 25.99/0.7740 | 26.01/0.6047 | 25.02/0.6004 | 26.13/0.8190 | 25.37/0.6878 |
| | ZS-N2S | 21.39/0.5390 | 17.46/0.4084 | 22.23/0.6428 | 21.83/0.5714 | 17.53/0.5771 | 25.14/0.6103 | 17.97/0.5082 | 24.33/0.7854 | 22.94/0.5959 | 21.15/0.7157 | 21.53/0.5467 | 19.43/0.6122 | 23.08/0.7725 | 21.23/0.6066 |
| | S2S | 16.58/0.5042 | 21.66/0.6523 | 18.07/0.6269 | 22.14/0.6206 | 15.01/0.5319 | 24.18/0.6548 | 18.73/0.6191 | 17.97/0.5682 | 24.78/0.7188 | 17.73/0.7807 | 21.51/0.6626 | 15.62/0.6042 | 22.09/0.7728 | 19.23/0.6553 |
| | **ZS-NCD** | 24.66/0.5807 | 27.63/0.7844 | 26.43/0.7443 | 26.40/0.7436 | 23.17/0.7781 | 27.53/0.6510 | 24.30/0.7680 | 29.58/0.8302 | 26.15/0.6997 | 26.46/0.7991 | 27.34/0.7196 | 26.68/0.7217 | 27.45/0.8440 | 26.44/0.7434 |
| 25 | BM3D | 20.61/0.4799 | 21.75/0.5241 | 22.93/0.6748 | 22.04/0.5620 | 19.94/0.6637 | 24.88/0.6358 | 22.35/0.7387 | 22.34/0.5167 | 23.11/0.5790 | 21.99/0.6799 | 22.75/0.5533 | 20.01/0.4239 | 23.48/0.7582 | 22.17/0.5992 |
| | JPEG2K | 22.59/0.5060 | 22.38/0.5732 | 22.81/0.5904 | 22.12/0.4455 | 19.59/0.6087 | 25.30/0.5533 | 21.51/0.6444 | 25.19/0.7343 | 22.10/0.4841 | 21.52/0.6226 | 24.78/0.6181 | 22.07/0.6515 | 22.44/0.7057 | 22.65/0.5952 |
| | DIP | 24.85/0.6243 | 24.02/0.5860 | 26.63/0.7646 | 25.41/0.6907 | 22.75/0.7640 | 27.42/0.6386 | 24.61/0.7935 | 29.42/0.8138 | 26.07/0.7119 | 27.02/0.8225 | 27.36/0.6839 | 26.90/0.7348 | 27.25/0.8467 | 26.13/0.7289 |
| | DD | 24.51/0.6017 | 23.30/0.6625 | 25.78/0.7389 | 24.96/0.6226 | 22.80/0.7680 | 28.18/0.6808 | 24.65/0.7872 | 30.34/0.8723 | 25.91/0.6824 | 26.31/0.8306 | 28.07/0.7242 | 26.92/0.7823 | 26.80/0.8309 | 26.04/0.7373 |
| | ZS-N2N | 25.81/0.6876 | 25.11/0.6387 | 27.17/0.7874 | 26.65/0.7562 | 23.45/0.7635 | 28.28/0.7277 | 24.95/0.8033 | 28.68/0.7031 | 28.10/0.7878 | 27.67/0.8276 | 27.47/0.6713 | 26.90/0.6864 | 27.43/0.8511 | 26.75/0.7455 |
| | ZS-N2S | 19.29/0.4409 | 21.36/0.5393 | 21.65/0.6879 | 20.44/0.4223 | 16.56/0.4988 | 25.41/0.6417 | 15.88/0.5870 | 25.29/0.8037 | 19.35/0.6591 | 21.09/0.6962 | 24.71/0.6801 | 22.47/0.7677 | 22.00/0.7806 | 21.19/0.6312 |
| | S2S | 17.40/0.5115 | 24.01/0.7441 | 18.56/0.6458 | 22.99/0.6967 | 15.48/0.5679 | 25.07/0.6826 | 19.74/0.7193 | 15.06/0.7385 | 26.12/0.7550 | 18.86/0.8094 | 23.26/0.6959 | 12.85/0.6302 | 22.89/0.8084 | 20.18/0.6927 |
| | **ZS-NCD** | 25.23/0.6145 | 29.09/0.8189 | 27.60/0.7969 | 27.66/0.8053 | 24.50/0.8223 | 28.31/0.6908 | 25.65/0.8231 | 30.99/0.8571 | 27.21/0.7491 | 27.82/0.8309 | 27.80/0.7490 | 28.42/0.8712 | 27.60/0.8823 | 27.60/0.7827 |
| 50 | BM3D | 21.78/0.5516 | 22.44/0.5350 | 23.85/0.7114 | 22.60/0.6103 | 21.36/0.7094 | 25.06/0.6670 | 23.51/0.7921 | 23.08/0.4721 | 23.86/0.6477 | 22.82/0.6871 | 23.24/0.5494 | 21.35/0.4511 | 24.07/0.7814 | 23.00/0.6281 |
| | JPEG2K | 23.83/0.5611 | 23.92/0.6612 | 24.26/0.6565 | 23.53/0.5775 | 21.09/0.6866 | 26.46/0.6041 | 22.89/0.7170 | 27.56/0.7817 | 23.77/0.5931 | 23.00/0.7123 | 26.46/0.6628 | 24.44/0.7320 | 24.00/0.7584 | 24.25/0.6696 |
| | DIP | 25.67/0.6702 | 26.62/0.6890 | 28.12/0.8173 | 27.79/0.8136 | 24.29/0.8131 | 28.63/0.7149 | 26.30/0.8449 | 30.97/0.8395 | 27.29/0.7409 | 29.10/0.8779 | 28.24/0.7107 | 28.64/0.7849 | 28.74/0.8816 | 27.72/0.7845 |
| | DD | 25.39/0.6439 | 23.57/0.6720 | 27.23/0.7930 | 26.69/0.7441 | 24.40/0.8285 | 28.69/0.7060 | 26.03/0.8358 | 32.04/0.8838 | 27.45/0.7575 | 27.94/0.8613 | 29.45/0.7641 | 29.37/0.8636 | 28.33/0.8740 | 27.43/0.7867 |
| | ZS-N2N | 27.06/0.7422 | 26.44/0.7151 | 29.33/0.8535 | 28.72/0.8361 | 25.47/0.8308 | 29.57/0.7822 | 26.98/0.8566 | 30.77/0.7918 | 30.02/0.8491 | 29.50/0.8730 | 29.03/0.7330 | 29.21/0.7853 | 29.34/0.8968 | 28.57/0.8112 |
| | ZS-N2S | 19.68/0.5011 | 19.80/0.4843 | 22.87/0.6603 | 19.48/0.3374 | 16.44/0.4783 | 18.21/0.4958 | 18.06/0.5203 | 25.22/0.8178 | 22.22/0.6118 | 21.95/0.7717 | 22.69/0.6847 | 19.84/0.7204 | 23.25/0.7586 | 20.75/0.6033 |
| | S2S | 19.15/0.5499 | 27.76/0.8445 | 19.57/0.6641 | 23.63/0.7330 | 16.26/0.6303 | 26.79/0.7083 | 20.99/0.7695 | 15.34/0.7890 | 27.87/0.8113 | 21.54/0.8363 | 26.51/0.7211 | 13.43/0.6657 | 23.93/0.8512 | 21.75/0.7365 |
| | **ZS-NCD** | 26.05/0.6622 | 30.15/0.8172 | 29.25/0.8571 | 29.83/0.8761 | 25.99/0.8641 | 29.36/0.7458 | 27.52/0.8790 | 32.51/0.8739 | 29.05/0.8147 | 29.27/0.8500 | 29.64/0.7647 | 29.53/0.7775 | 30.03/0.9081 | 29.09/0.8223 |

## C.3 Kodak24 Dataset

For Gaussian denoising $\lambda = (75, 150, 750)$ for noise levels $\sigma_z = (15, 25, 50)$ and for Poisson denoising $\lambda = (750, 300, 150)$ for $\alpha = (15, 25, 50)$. For BM3D Poisson denoising of $\alpha = (15, 25, 50)$ we set $\sigma_{\mathrm{BM3D}} = (50, 25, 15)$. We report the detailed results of AWGN denoising in Table 12, and Poisson noise denoising in Table 13.

Table 12: Kodak24 Denoising performance comparison under AWGN denoising $\mathcal{N}(0, \sigma_z^2 I)$.

Table 13: Kodak24 Denoising performance performance under Poisson noise $\text{Poisson}(\alpha\boldsymbol{x})/\alpha$.

| Method ($\alpha$) | 01 | 02 | 03 | 04 | 05 | 06 | 07 | 08 | 09 | 10 | 11 | 12 | 13 | 14 | 15 | 16 | 17 | 18 | 19 | 20 | 21 | 22 | 23 | 24 | Average |
|---|---|---|---|---|---|---|---|---|---|---|---|---|---|---|---|---|---|---|---|---|---|---|---|---|---|

## C.4 Microscopy Mouse Nuclei Dataset

For these images with noise level $\sigma_z = (10, 20)$ we set $\lambda = (200, 600)$, we train the networks for 20K steps to obtain the results. We report the detailed denoising performance in Table 14 and 15 respectively.

Table 14: Denoising performance under AWGN $\mathcal{N}(0, \sigma^2 I)$ on fluorescence microscopy dataset: Mouse Nuclei. Images are cropped into $128 \times 128$. Noise level $\sigma_z = 10$.

| # | JPEG2K | BM3D | DIP | DD | ZS-N2N | ZS-N2S | S2S | ZS-NCD |
|---|---|---|---|---|---|---|---|---|
| 1 | 32.90/0.7954 | 38.88/0.9631 | 37.31/0.8973 | 37.73/0.9464 | 36.37/0.9356 | 34.70/0.9410 | 10.88/0.1687 | 39.03/0.9556 |
| 2 | 32.32/0.8300 | 37.53/0.9613 | 35.93/0.8909 | 36.46/0.9560 | 35.26/0.9345 | 28.78/0.8504 | 13.08/0.4000 | 36.83/0.9546 |
| 3 | 32.97/0.8584 | 38.43/0.9690 | 36.17/0.8482 | 37.03/0.9631 | 35.86/0.9405 | 31.53/0.9307 | 12.76/0.3374 | 37.81/0.9634 |
| 4 | 32.57/0.8418 | 38.05/0.9605 | 35.82/0.9107 | 36.70/0.9478 | 34.86/0.9066 | 32.13/0.8688 | 14.42/0.3639 | 37.51/0.9303 |
| 5 | 34.54/0.7646 | 41.53/0.9596 | 38.09/0.8268 | 40.02/0.9438 | 39.30/0.9252 | 29.75/0.7976 | 10.22/0.1165 | 40.93/0.9420 |
| 6 | 32.02/0.8860 | 37.49/0.9703 | 35.24/0.8997 | 36.05/0.9628 | 35.38/0.9491 | 30.63/0.8989 | 14.42/0.3931 | 37.26/0.9588 |
| Average | 32.89/0.8294 | **38.65/0.9640** | 36.43/0.8789 | 37.33/0.9533 | 36.17/0.9319 | 31.26/0.8812 | 12.63/0.2966 | 38.23/0.9508 |

Table 15: Denoising performance under AWGN $\mathcal{N}(0, \sigma^2 I)$ on fluorescence microscopy dataset: Mouse Nuclei. Images are cropped into $128 \times 128$. Noise level $\sigma_z = 20$.

| # | JPEG2K | BM3D | DIP | DD | ZS-N2N | ZS-N2S | S2S | ZS-NCD |
|---|---|---|---|---|---|---|---|---|
| 1 | 28.37/0.6337 | 35.10/0.9211 | 33.09/0.8485 | 33.70/0.8938 | 32.32/0.8609 | 32.59/0.8763 | 9.30/0.0240 | 34.98/0.8843 |
| 2 | 27.97/0.7255 | 33.80/0.9410 | 31.41/0.7986 | 32.39/0.9239 | 31.07/0.8421 | 28.42/0.8328 | 10.73/0.2383 | 33.75/0.9172 |
| 3 | 28.42/0.7121 | 34.45/0.9352 | 31.47/0.7642 | 32.64/0.9096 | 31.63/0.8660 | 31.08/0.9003 | 10.12/0.1807 | 34.25/0.9216 |
| 4 | 29.31/0.7557 | 34.30/0.9245 | 31.02/0.7598 | 32.71/0.9008 | 31.12/0.8168 | 30.60/0.8551 | 11.33/0.1947 | 33.87/0.8947 |
| 5 | 29.62/0.5932 | 38.50/0.9158 | 35.45/0.7763 | 37.18/0.9068 | 35.90/0.8585 | 32.89/0.8640 | 8.23/0.0650 | 37.70/0.9137 |
| 6 | 27.71/0.7713 | 33.61/0.9399 | 31.48/0.7860 | 32.41/0.9206 | 31.43/0.8750 | 26.87/0.8312 | 10.83/0.2328 | 33.72/0.9245 |
| Average | 28.57/0.6986 | **34.96/0.9296** | 32.32/0.7889 | 33.50/0.9092 | 32.25/0.8532 | 30.41/0.8600 | 10.09/0.1559 | 34.71/0.9093 |

## C.5 Real Camera Noise Dataset PolyU

For these images with unknown noise model/level $\lambda = 25$. Also for BM3D the best peroformance was achieved with setting $\sigma_{\text{BM3D}} = 15$. We report the detailed denoising performance in Table 16.

Table 16: Real camera denoising performance on camera image dataset: PolyU. The dataset includes photos taken from 3 brands of cameras. Randomly selected 6 images are cropped into $512 \times 512$.

| Models | C.plug11 | C.bike10 | N.flower1 | N.plant10 | S.plant13 | S.door10 | Average |
|--------|----------|----------|-----------|-----------|-----------|----------|---------|
| JPEG2K | 36.26 / 0.9615 | 34.23 / 0.9371 | 33.55 / 0.9194 | 36.74 / 0.9157 | 30.39 / 0.9001 | 34.84 / 0.9012 | 34.33 / 0.9225 |
| BM3D | 37.15 / 0.9758 | **34.85 / 0.9615** | **35.81** / 0.9504 | 38.40 / 0.9410 | 31.65 / 0.9465 | 36.43 / 0.9285 | 35.71 / 0.9506 |
| DIP | 37.62 / 0.9724 | 34.85 / 0.9534 | 34.93 / 0.9396 | 37.64 / 0.9256 | 31.50 / 0.9396 | 36.02 / 0.9145 | 35.43 / 0.9408 |
| DD | 36.79 / 0.9722 | 34.73 / 0.9566 | 34.85 / 0.9366 | 37.84 / 0.9327 | 30.91 / 0.9305 | 33.88 / 0.9084 | 34.83 / 0.9395 |
| ZS-N2N | 36.30 / 0.9621 | 33.18 / 0.8853 | 33.28 / 0.8974 | 36.21 / 0.8862 | 30.57 / 0.9052 | 34.89 / 0.8804 | 34.07 / 0.9028 |
| ZS-N2S | 22.76 / 0.9119 | 20.36 / 0.8133 | 25.20 / 0.8670 | 33.63 / 0.8920 | 21.33 / 0.8256 | 18.39 / 0.6966 | 23.61 / 0.8344 |
| S2S | **37.75 / 0.9765** | 33.56 / 0.9545 | 35.78 / **0.9537** | 38.30 / 0.9398 | **31.93 / 0.9483** | 36.65 / 0.9433 | 35.66 / 0.9527 |
| ZS-NCD | 36.99 / 0.9763 | 34.79 / 0.9586 | 35.43 / 0.9489 | **38.65 / 0.9449** | 31.79 / 0.9464 | **37.42 / 0.9451** | **35.84 / 0.9534** |

# D Visual Comparisons

In this section, we provide more visualization comparison of the zero-shot denoisers. The reconstruction PSNR and SSIM are above the images.

## D.1 Kodak24

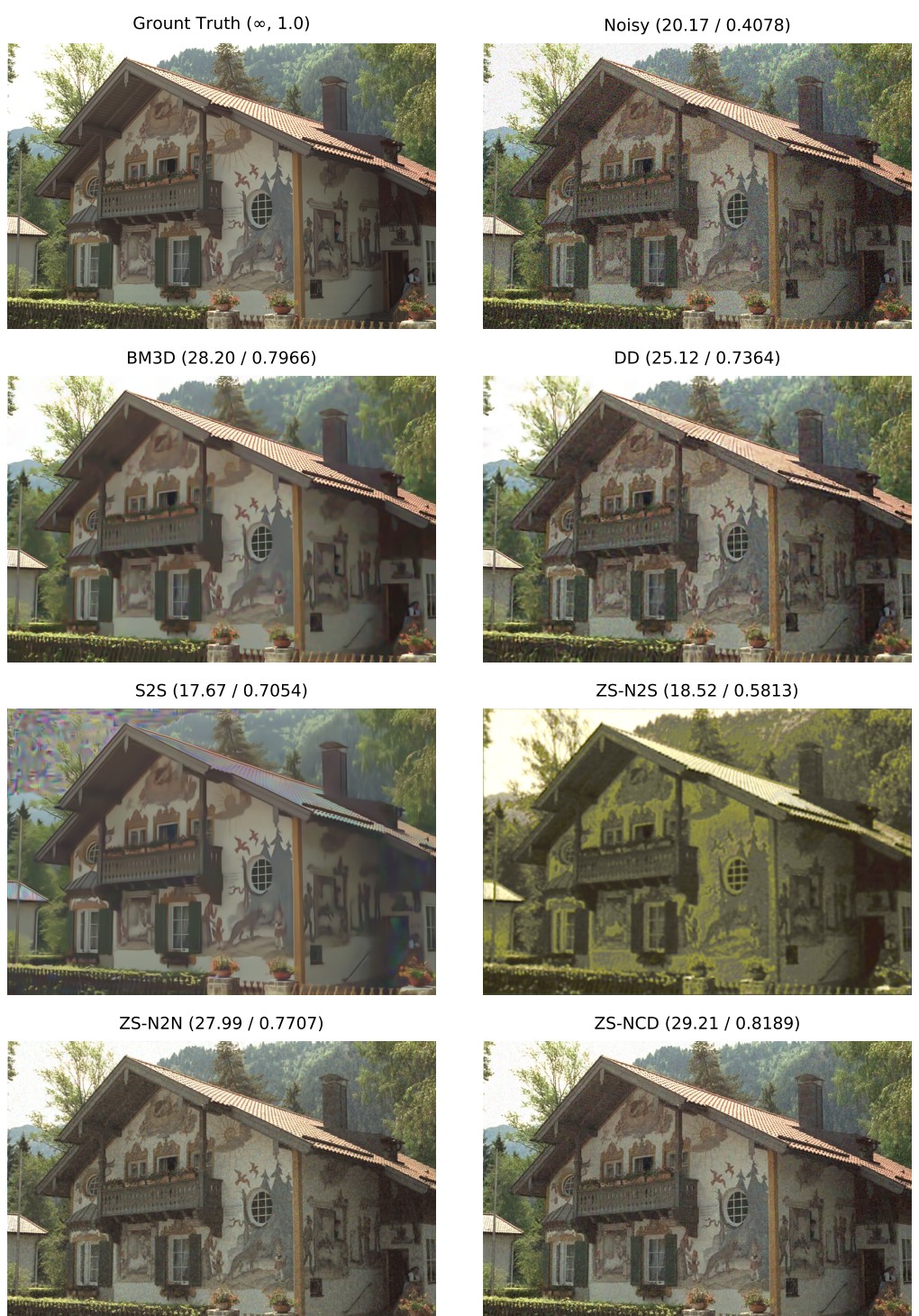

Figure 7: *Kodim24* under additve white Gaussian noise ($\sigma_z = 25$).

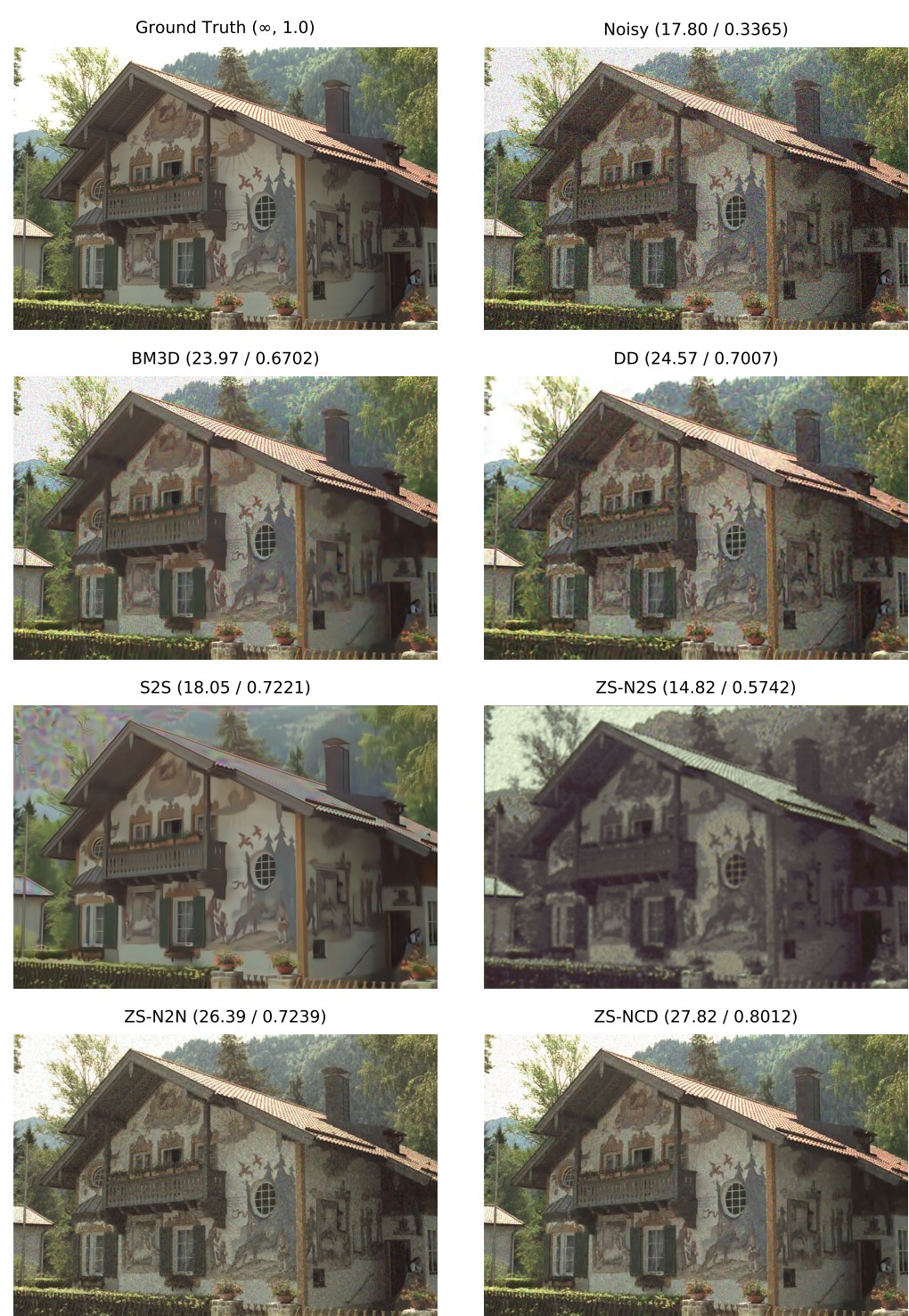

Figure 8: *Kodim24* under Poisson noise ($\alpha = 25$).

## D.2 Mouse Nuclei

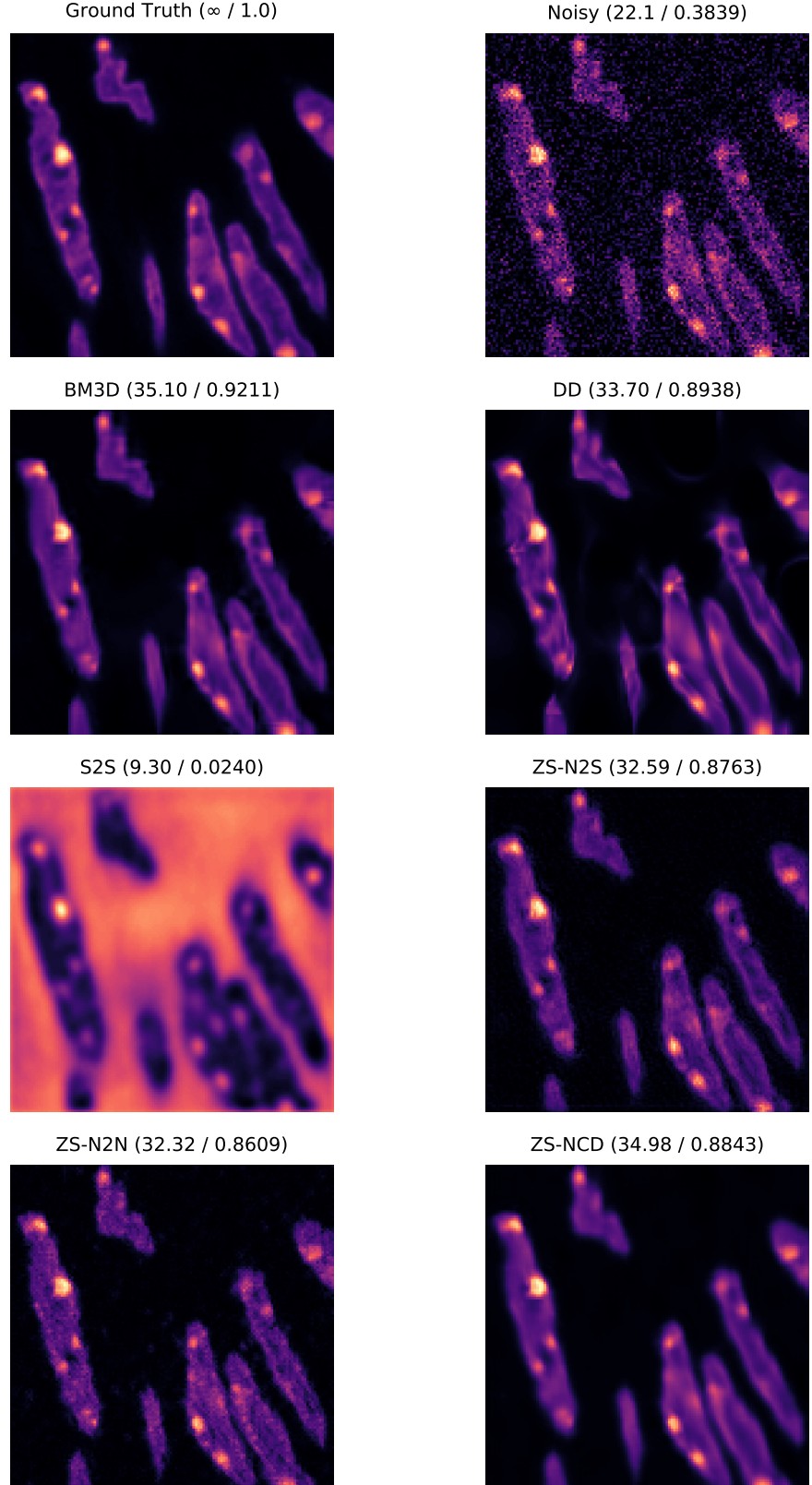

Figure 9: Mouse nuclei reconstruction comparison under additive white Gaussian noise ($\sigma_z = 20$).

