# OpenReview forum: "Zero-shot Denoising via Neural Compression: Theoretical and algorithmic framework"
_NeurIPS.cc/2025/Conference — NeurIPS 2025 spotlight_

### Official Review · Reviewer_TJvX · 2025-06-29

**Clarity:** 3
**Significance:** 4
**Originality:** 3
**Rating:** 5
**Confidence:** 4

**Summary:**

The authors propose a novel method for zero-shot image denoising via neural compression in the single-image setting. Specifically, given a single noisy image, an encoder-decoder neural network with a rate constraint is trained to compress and decompress overlapping patches extracted from the image. Once trained, the denoised image is reconstructed by applying the compression-decompression process to all patches and aggregating the outputs. The approach builds on information-theoretic principles, assuming that clean images are more structured and thus more compressible than noisy ones. The rate constraint naturally biases the model toward reconstructing structured (clean) components while suppressing unstructured (noisy) variations.

**Questions:**

I do not have questions for the authors.

**Ethical Concerns:**

["NO or VERY MINOR ethics concerns only"]

**Final Justification:**

I have no concerns about this paper.

**Limitations:**

The authors address the limitations in section 6.

**Paper Formatting Concerns:**

I do not have formatting concerns.

**Quality:**

3

**Strengths And Weaknesses:**

The proposed method is well-justified theoretically and backed by finite-sample performance guarantees for both AWGN and Poisson noise models. Extensive experiments are conducted on diverse image domains (natural, microscopy, real-world camera images) and across multiple noise types. The results are consistently strong and demonstrate state-of-the-art performance among zero-shot methods. The architecture and optimization procedure are well-detailed, and the authors provide reproducibility information.

---

> ### Author Rebuttal · Authors · 2025-07-30
>
> We appreciate the reviewer’s positive feedback.

---

### Official Review · Reviewer_FvQP · 2025-06-29

**Clarity:** 3
**Significance:** 2
**Originality:** 2
**Rating:** 6
**Confidence:** 3

**Summary:**

This paper is composed of two parts. The first part focuses on the relationship between compression and denoising, and presents several theoretical results that give upper bounds to the denoising performance of a compression-based denoiser. The second part applies compression-based denoising to the "zero-shot" setting, where only a single noisy image is available. The original high-resolution image is broken up in small patches that are used to train a neural compressor. Numerical experiments demonstrate the method performs comparably to other zero-shot baselines.

**Questions:**

- Do the authors expect the dependence of their bounds in rate $R$ and distortion $\delta$ to be tight?
- The authors propose to select the RD tradeoff parameter $\lambda$ such that the distortion of the compressed noisy patches becomes comparable to the noise level. Another natural choice would be to optimize the upper bounds given in the Theorems along the RD curve. For instance, Theorem 1 suggests to set $\lambda = (8\ln 2)(1 + 2\sqrt \eta)^2\frac{\sigma_z^2}{n}$. How do these two criteria compare?
- More broadly, I am surprised that "compression performs better at denoising than denoising". That is, why does compressing 8x8 patches would outperform an approach tailored to denoising such as SURE or any other mentioned in the paper? The authors mention an increased robustness to overfitting, but it is not intuitive to me why a compressor would overfit less than a denoiser (e.g., in SURE, the divergence of the denoiser is explicitly regularized).

**Ethical Concerns:**

["NO or VERY MINOR ethics concerns only"]

**Final Justification:**

I have no remaining concerns.

**Limitations:**

yes

**Quality:**

3

**Strengths And Weaknesses:**

## Compression-based denoising

The paper makes novel contributions to compression-based denoising through Theorems 1 to 3, which show that better compression performance (as measured by rate and distortion) leads to an improved upper bound on denoising performance. However, it is in my opinion difficult to evaluate these results due to the choice of setting. Indeed, denoising performance is here measured in the worst case over a deterministic class of signals, rather than average MSE over a signal distribution as is more common. In particular, this is the metric that is used in the numerical comparisons of Table 1. Thus, it is unclear if additional factors compared to optimal MMSE performance (such as $\log_2 n$ in Corollary 1) are due to this choice of setting, looseness of the bounds, or a fundamental underperformance of compression-based denoising.

Related to the above point, the authors propose to select the codeword $c$ via a maximum a posteriori procedure, rather than minimum mean squared error (eqs. (2) vs (3)). Although it is stated that the former is always the best choice, it seems to me that the upper bound in Theorem 3 becomes more advantageous to that of Theorem 2 in the regime of small rates and large distortions.

## Application to zero-shot denoising

A strength of the paper is its experimental results, which show superior zero-shot denoising performance to other approaches. I note that there are other very recent concurrent works, which build on ZS-N2N and may be relevant to this paper:

> Bai, J., Zhu, D., & Chen, M. (2025). Dual-sampling noise2noise: Efficient single image denoising. IEEE Transactions on Instrumentation and Measurement.
>
> Ma, Q., Jiang, J., Zhou, X., Liang, P., Liu, X., & Ma, J. (2025). Pixel2Pixel: A Pixelwise Approach for Zero-Shot Single Image Denoising. IEEE Transactions on Pattern Analysis and Machine Intelligence.

As defined by NeurIPS policy for concurrent works, these works do not qualify as prior work, but could be of interest to the authors.

A critical component of the method is the chosen patch size. I could not find it mentioned in the paper (though inspection of the figure indicates that it might be $8 \times 8$?). How was it selected? It seems to me that different choices of patch sizes would greatly affect the results. For instance, a small patch size limits the range of spatial dependencies in the signal that can be exploited by the denoiser, effectively limiting it to small noise levels. However, a large patch size leads to a small number of patches available to train the compressor.

---

> ### Author Rebuttal · Authors · 2025-07-30
>
> We thank the reviewer for their valuable feedback. We first address the specific questions raised, followed by a response to the points noted in the weaknesses section. In our response to Question 1, we also address some of the comments regarding the choice of settings in the theoretical results, as mentioned in the Strengths And Weaknesses.
>
> >Q1. Tightness of bounds in Section 3.
>
> Our analysis is framed in a deterministic, high-probability setting: we provide uniform high-probability guarantees on reconstruction error for signals in a fixed class, without assuming any prior distribution. This framework is common in the information-theoretic literature on compression-based inference (e.g., compression-based compressed sensing), and it offers practical advantages, particularly in the zero-shot denoising scenario where the underlying data distribution may be unknown or difficult to model. The deterministic nature of our bounds ensures they hold uniformly over the signal class, rather than in expectation.
>
> Regarding the tightness of the bounds, in the discussion following Corollary 1, we have compared our result with the optimal Bayesian performance (in terms of information dimension). In the revised version, we will also include a comparison to known minimax rates for $k$-sparse signal recovery under Gaussian noise. For the case $k=k_n$, such that $k_n/n\to 0$, as $n$ grows without bound,  the minimax  risk in known to scale as ${\sigma_z^2 k \log(n/k) \over n} $ [Johnstone2019] [Donoho1994] [Donoho1992]. This further confirms the tightness of our bound in the worst-case setting, and potential optimality of compression-based compressed sensing.
>
> Finally, while in Table 1 we report average denoising performance, our method is designed to perform well on each individual instance, consistent with the worst-case theoretical guarantees we provide. This aligns with the goals of zero-shot denoising, where robustness to the specific realization is more critical than expected-case optimality.
>
> [Johnstone2019] Johnstone, I. M. Gaussian Estimation: Sequence and Wavelet Models.
>
> [Donoho1994] Donoho, D. L., et al. Minimax risk over $\ell_p$ balls for $\ell_q$ losses.
>
> [Donoho1992] Donoho, D. L., et al. Maximum entropy and the nearly black object.
>
> >Q2. Selecting $\lambda$ based on theoretical upper bounds.
>
> Theorem 1  provides a non-asymptotic upper bound on the reconstruction error, assuming that a compression code with a given rate $R$ and distortion $\delta$ is available. The constants appearing in the bound (such as $(8 \ln 2)(1 + 2\sqrt{\eta})^2$) characterize the relationship between rate and MSE, but they are not intended to prescribe a specific choice of the Lagrange multiplier $\lambda$ used in training.
>
> Indeed, in principle, one could consider minimizing the upper bound in Theorem 1 as a function of $\delta$ (or $R$), which in turn could suggest an “optimal” rate-distortion operating point for denoising. However, solving such an optimization is highly non-trivial. Specifically, note that the upper bound in Theorem 1 can be written as
> $$
> u(\delta) = \sqrt{\delta} + \sigma_z \zeta \sqrt{R(\delta)},
> $$where $\zeta = 2 \sqrt{\frac{2 \ln 2}{n}}(1 + 2\sqrt{\eta})$, and $R(\delta)$ is the rate corresponding to distortion $\delta$. Differentiating and setting $u'(\delta) = 0$ gives:
> $$
> \frac{1}{2\sqrt{\delta}} + \sigma_z \zeta \cdot \frac{R'(\delta)}{2\sqrt{R(\delta)}} = 0.
> $$However, since the function $R(\delta)$ is generally not known in closed form for real-world data distributions or learned compressors, this optimization cannot be carried out analytically or even tractably in practice.
>
> In contrast, during training, $\lambda$ is used as a Lagrange multiplier in a rate-distortion objective to implicitly select an operating point on the empirical RD curve of the single image. This RD curve is not analytically accessible and varies from image to image, making the mapping from $\lambda$ to $(R, \delta)$ nontrivial and data-dependent.
>
> Our proposed heuristic, choosing $\lambda$ such that the distortion roughly matches the noise level, is based on signal recovery intuition and performs well empirically, as demonstrated in our experiments and in Section 4, "*Setting the hyperparameter $\lambda$*".
>
> >Q3. Seemingly surprising effectiveness of compression at denoising.
>
> This is an insightful and intuitive question. While it may seem surprising that a compressor, which is not explicitly trained to denoise, can outperform methods designed specifically for denoising, the idea of using compression codes to solve inverse problems such as denoising and compressed sensing has a long history in information theory and signal processing. Prior work has shown that lossy compression codes can be used as a powerful implicit prior for solving inverse problems (see, e.g., [Donoho2002, Weissman2005, Jalali2016, Rezagah2017, Chang1997, Chang2000, Natarajan2002]).
>
> In the zero-shot setting, where only a single noisy observation is available and no clean data is accessible, most learning-based denoisers rely on indirect forms of regularization to prevent overfitting. These include masking strategies (ZS-N2S), architectural bias (DIP), or underparameterized networks (Deep Decoder). While useful, these techniques are often heuristic and do not provide a principled way to control information flow from input to output.
>
> In contrast, compression-based denoising  imposes a fundamental information-theoretic constraint via the rate-distortion trade-off. This restricts the number of bits that can pass through the model's bottleneck, acting as a principled regularizer that prevents the model from memorizing noise. The model is therefore implicitly biased toward compressible (structured) reconstructions, which aligns well with the nature of clean signals.
>
> Although SURE provides an unbiased estimate of the MSE under Gaussian noise with known variance, its effectiveness depends on accurate noise modeling and stable estimation of the divergence term. In practice, especially in the zero-shot setting with a single noisy image and no access to clean data, these conditions may not hold. Moreover, SURE implementations often rely on Monte Carlo approximations of the divergence, which can introduce significant variance, particularly in deep neural networks.
>
> Thus, while traditional denoising methods are explicitly designed for the task, compression-based approaches like ZS-NCD offer a robust and principled alternative in data-scarce regimes, combining inductive bias with effective optimization.
>
> [Chang1997] Chang, S. G., et al. Image denoising via lossy compression and wavelet thresholding.
>
> [Chang2000] Chang, S. G., et al. Adaptive wavelet thresholding for image denoising and compression.
>
> [Donoho2002] Donoho, D. L. The kolmogorov sampler.
>
> [Natarajan2002] Natarajan, B. K. Filtering random noise from deterministic signals via data compression.
>
> [Weissman2005] Weissman, T., et al. The empirical distribution of rate-constrained source codes.
>
> [Jalali2016] Jalali, S., et al. From compression to compressed sensing.
>
> [Rezagah2017] Rezagah, F. E., et al. Compression-based compressed sensing.
>
> [Soltanayev2018] Soltanayev, S., et al. Training deep learning based denoisers without ground truth data.
>
> [Metzler2018] Metzler, C. A., et al. Unsupervised learning with stein's unbiased risk estimator.
>
> >W1. Better bound in Theorem 3 over Theorem 2.
>
> Note that while the bound in Theorem 2 is on MSE, the bound in Theorem 3 is on RMSE. To compare  the results of the two theorems, we can square the bound in Theorem 3, which shows that while the MSE in Theorem 2 scales as $\sqrt{R/n}$, the MSE in Theorem 3 scales as $R^2/n$. This suggests  that, unlike in Theorem 2, for the bound in Theorem 3 to be small,  the rate $R$ cannot scale linearly with $n$. Therefore, minimizing the log-likelihood over a compression code should, in principle, yield better performance under Poisson noise. However, due to the highly non-convex nature of the log-likelihood loss, minimizing the MSE loss may lead to better performance in practice. A brief discussion of this observation, along with empirical results, is provided in Appendix B.2.
>
> >W2. Two concurrent zero-shot denoisers.
>
> We thank the reviewer for bringing these results to our attention.  The recent works advance the ZS-N2N framework. We will cite both references in the Related Work Section (Section 2) of our manuscript. Additionally, we have tested DS-N2N and Pixel2Pixel on the Kodak24 dataset under AWGN using the public code provided by the authors. Average PSNR/SSIM over the 24 images are reported in the table below.
>
> ||$\sigma=15$| $\sigma=25$| $\sigma=50$|
> |:-:|:-:|:-:|:-:|
> |DS-N2N|32.31 / 0.8803|29.64 / 0.8044|25.42 / 0.6378|
> |Pixel2Pixel|31.31 / 0.8707| 29.89 / 0.8098|26.55 / 0.6873|
> |ZS-NCD (ours)|**33.18 / 0.9026**|**30.60 / 0.8144**|**27.89 / 0.7464**|
>
> [Bai2025] **DS-N2N**: Bai, J., et al. Dual-sampling noise2noise: Efficient single image denoising.
>
> [Ma2025] **Pixel2Pixel**: Ma, Q., et al. Pixel2Pixel: A Pixelwise Approach for Zero-Shot Single Image Denoising.
>
> >W3. Patch size in ZS-NCD.
>
> We thank the reviewer for pointing this out. We indeed use a patch size of $8 \times 8$ in all experiments and will clearly state this in the revised version of the paper.
>
> Empirically, we observe that ZS-NCD is fairly robust to patch size, as long as the size is within a reasonable range, i.e., large enough to contain meaningful structure, but small enough to ensure enough training data per image.
>
> To support this, we include below denoising results of ZS-NCD for different patch sizes, reported as PSNR (dB)/SSIM, on the *kodim05* image from the Kodak24 dataset under AWGN.
>
> |$\sigma$| 4x4 | 8x8 | 16x16 | 32x32|
> |:-:|:-:|:-:|:-:|:-:|
> |15|31.76 / 0.9246|31.63 / 0.9286|31.66 / 0.9285| 31.70 / 0.9280|
> |25|28.46 / 0.8439|28.81 / 0.8615|29.01 / 0.8647|28.92 / 0.8621|
> |50|24.79 / 0.7264|25.60 / 0.7608| 25.58 / 0.7608|25.60 / 0.7613|

---

> > ### Comment · Reviewer_FvQP · 2025-08-04
> >
> > Thank you for your detailed reply, which has addressed my questions. It is interesting to see that indeed the optimal patch size increases with the noise level. It could be interesting in future work to study this further.
> >
> > I will increase my score. This paper makes valuable theoretical and practical contributions to the problem of zero-shot denoising.

---

### Official Review · Reviewer_t45e · 2025-07-01

**Clarity:** 3
**Significance:** 2
**Originality:** 2
**Rating:** 4
**Confidence:** 3

**Summary:**

**Focus Issue**


Zero-shot denoising: Denoising observations without training samples or clean reference images.


**Contributions**


The paper proposes the Zero-Shot Neural Compression Denoiser (ZS-NCD):1.It optimizes on patches extracted from a single noisy image. The final reconstruction is obtained by aggregating outputs from the trained model over overlapping patches.
2.It provides novel finite-sample theoretical results describing the achievable upper bound on reconstruction error for maximum likelihood-based compression denoisers, further establishing the theoretical foundation of compression-based denoising.


**Performance**

ZS-NCD naturally avoids overfitting and eliminates the need for manual regularization or early stopping.
ZS-NCD achieves state-of-the-art performance among zero-shot denoisers for both Gaussian and Poisson noise removal tasks.

**Questions:**

In addition to the doubts raised in the "Strengths And Weaknesses" section, the article does not present the performance of the proposed method on completely clean images. Specifically, ZS-NCD achieves the desired results by overlapping and training patches from the same image. This raises the following questions:

Will it significantly degrade the quality of clean images?


Is the computational cost acceptable since a model has to be trained for each individual image?


Does a model trained on a specific image possess generalizability? If not, would it be impossible to complete the denoising task for a large number of images within a reasonable timeframe?


If the authors can address these questions, I would consider raising the score.

**Ethical Concerns:**

["NO or VERY MINOR ethics concerns only"]

**Final Justification:**

We still have some unresolved concerns, and the authors have not provided further clarification. Nevertheless, we believe that this method represents the current state-of-the-art (SOTA), and some concerns are inherent issues within the field. Therefore, we have decided to give a score of 4.

**Limitations:**

Yes

**Paper Formatting Concerns:**

The text in images similar to Figure 6 is too small.

The appendix simply lists images without adequate elaboration.

**Quality:**

3

**Strengths And Weaknesses:**

**Quality and Clarity**


The presentation of key issues and solutions is relatively clear, and the figures and tables are displayed in a neat and concise manner. Therefore, I tend to give a positive rating in terms of Quality and Clarity.


**Significance and Originality**


The proposed problem, "Zero-shot denoising via neural compression," is valuable, and its innovativeness is also recognizable. However, the baseline models used for comparison are somewhat outdated, with most dating back to 2020 or earlier. Additionally, I do not see a significant leading edge in the performance metrics. Therefore, the claim that "S-NCD achieves state-of-the-art performance among zero-shot denoisers for both Gaussian and Poisson noise removal tasks" fails to convince me. Furthermore, the experimental discussion regarding "naturally avoiding overfitting and eliminating the need for manual regularization or early stopping" is overly simplistic and almost glossed over. In summary, I tend to give a relatively conservative rating.

---

> ### Author Rebuttal · Authors · 2025-07-30
>
> We thank the reviewer for their valuable feedback. We first address the specific questions raised, followed by a response to the points noted in the weaknesses section.
>
> >Q1. Will it significantly degrade the quality of clean images?
>
> ZS-NCD learns a lossy compression code using the patches extracted from the input noisy image. When the input is a noisy image, enforcing a rate-distortion trade-off via a penalty  on the rate  effectively suppresses noise and yield a denoised image. The denoising performance depends on the rate constraint. To achieve the best performance, one needs to set the rate penalty such that the operating rate-distortion point is consistent with the noise level.  If the input image is a clean noise-free image, the output may or may not be distorted depending on the rate penalty. In the case of no rate penalty ($\lambda = 0$), ZS-NCD is able to recover an almost lossless reconstruction of the clean image, provided the network bottleneck has sufficient capacity for the required entropy (as distortion goes to zero, required rate grows without bound).
>
> This contrasts with methods like ZS-N2N, ZS-N2S and S2S, which  inherently cannot  recover a clean image due to disjoint-pixel training strategies or masking.
>
> To further clarify this point, we empirically evaluated different methods on a clean image (*kodim05* from Kodak24 dataset). The results, reported as PSNR (dB)/SSIM, are shown in the following table. $n_b$ denotes the dimension of latent code in the bottleneck of ZS-NCD.
>
> || kodim05 ($\sigma=0$)|
> |:----------:|:------:|
> |DIP  | 42.02 / 0.9913|
> |DD  |28.14 / 0.8736|
> |ZS-N2N  | 33.46 / 0.9845|
> |ZS-NCD ($n_b=32$) | 35.52 / 0.9797
> |ZS-NCD ($n_b=128$) | 54.57 / 0.9997
>
>
> >Q2. Is the computational cost acceptable since a model has to be trained for each individual image?
>
> Training a model per noisy image is an inherent aspect of zero-shot learning-based denoisers, which assume no access to training datasets or noisy/clean image pairs. This is true for all zero-shot learning-based denoisers, and not just ZS-NCD.  This implies that compared to semi-supervised and supervised methods, such algorithms require a higher computational complexity during test time. However, the trade-off is justified in cases where access to data is very limited. This is true in various biomedical imaging  and scientific imaging applications.
>
> We provide timing and hardware details  in Appendix C (lines 638–643). Note that ZS-NCD has not yet been optimized for speed. We believe that there is considerable room for reducing the training time through more efficient architecture designs, weight reuse and adaptive training. The goal of this work is to both empirically and theoretically demonstrate the promise of compression-based zero-shot learning-based denoising, as a novel framework.
>
> >Q3. Does a model trained on a specific image possess generalizability? If not, would it be impossible to complete the denoising task for a large number of images within a reasonable timeframe?
>
> This is an interesting question. The notion of generalizability is different in  zero-shot learning-based denoisers  compared to  supervised methods. In zero-shot settings, the model is trained and evaluated on a single noisy image. Therefore, generalization beyond the observed image is not a goal. However, exploring generalization across images is an intriguing question. Zero-shot denoising methods, such as DIP and deep decoder, are image-dependent and fail completely when applied to an unseen noisy image. On the other hand, ZS-NCD, due to its compression-based structure and the fact that it is trained on image patches,  shows promising generalizability  when trained on a noisy image and applied to another unseen noisy image.
>
> To empirically verify this point, in the following two tables, we report the performance of ZS-NCD trained on a noisy image and used to denoise another image, for two different noise levels ($\sigma=15$ and $\sigma=50$). In each case, we also report the performance achieved when ZS-NCD is used in its original form, i.e., using the same training and test image. It can be observed that, despite expected performance drop, still ZS-NCD shows reasonable and promising generalization. *kodim01* and *kodim05* are chosen from Kodak24 dataset.
>
> |AWGN ($\sigma=15$) | Test on kodim01 | Test on kodim05|
> |:----------:|:------:|:------:|
> |**Train on kodim01** | 31.28 / 0.9059 | 26.71 / 0.8944|
> |**Train on kodim05** | 30.29 / 0.8874 | 31.63 / 0.9286|
>
> |AWGN ($\sigma=50$) | Test on kodim01 | Test on kodim05|
> |:----------:|:------:|:------:|
> |**Train on kodim01** | 26.01 / 0.7245 | 23.59 / 0.7300|
> |**Train on kodim05** | 25.19 / 0.6966 | 25.60 / 0.7608|
>
>
> >W1. More recent baselines comparison.
>
> The most recent state-of-the-art learning-based zero-shot denoiser we were aware of at the time of submission was ZS-N2N [Mansour2023], published in 2023. ZS-N2N is included in our comparisons as reported for example in Table 1 of the manuscript. We have since become aware of two concurrent zero-shot denoising methods:  DS-N2N [Bai2025] and Pixel2Pixel [Ma2025], pointed to us by Reviewer FvQP. Both algorithms are variations of ZS-N2N [Mansour2023].
>
> In the revised version of our manuscript we will cite both papers in the Related Work section.
> Additionally, we have tested DS-N2N and Pixel2Pixel on the Kodak24 dataset using the public code provided by the authors on our own machines, in line with our policy of reproducing all denoisers performances under the same condition for fairness.
> In the table below, we compare the performance of our proposed ZS-NCD under AWGN. The table reports the average PSNR (dB) and SSIM over the 24 images in Kodak24 dataset.
>
> || $\sigma=15$ | $\sigma=25$ | $\sigma=50$|
> |:---|:---|:---|:---|
> |ZS-N2N (2023) |  32.30 / 0.8650 | 29.54 / 0.7798  | 25.82 / 0.6151|
> |DS-N2N (2025) | 32.31 / 0.8803 | 29.64 / 0.8044  | 25.42 / 0.6378|
> |Pixel2Pixel (2025) | 31.31 / 0.8707  | 29.89 / 0.8098  | 26.55 / 0.6873|
> |ZS-NCD (ours) | **33.18 / 0.9026**  | **30.60 / 0.8144** | **27.89 / 0.7464** |
>
> [Mansour2023] **ZS-N2N**: Mansour, Y., \& Heckel, R. (2023). Zero-shot noise2noise: Efficient image denoising without any data. In Proceedings of the IEEE/CVF Conference on Computer Vision and Pattern Recognition (pp. 14018-14027).
>
> [Bai2025] **DS-N2N**: Bai, J., Zhu, D., \& Chen, M. (2025). Dual-sampling noise2noise: Efficient single image denoising. IEEE Transactions on Instrumentation and Measurement.
>
> [Ma2025] **Pixel2Pixel**: Ma, Q., Jiang, J., Zhou, X., Liang, P., Liu, X., \& Ma, J. (2025). Pixel2Pixel: A Pixelwise Approach for Zero-Shot Single Image Denoising. IEEE Transactions on Pattern Analysis and Machine Intelligence.
>
> >W2. Clarification on performance claims  of ZS-NCD.
>
> The claim refers to ZS-NCD's performance  among **learning-based zero-shot denoisers**. To ensure that the wording of the claim is clear, in the revised version, we will emphasize that the comparison is within learning-based zero-shot denoising methods. Specifically, we will update that sentence as follows: **ZS-NCD achieves state-of-the performance among learning-based zero-shot denoisers for both Gaussian and Poisson noise**, as supported by the empirical  results reported in Table 1 and Table 2 of the manuscript. In both tables, we have included the performance of BM3D as a strong classical non-learning-based baseline to provide broader context.
>
> >W3. Clarification on overfitting claims.
>
> The statement in the abstract is further elaborated  in Section 5, ``*Robustness to Overfitting*'' (lines 282–290) and is empirically supported by Figure 2. Specifically, ZS-NCD leverages the entropy constraint of the  compression-based architecture, which   inherently limits amount of information that can pass through the bottleneck. As a result, the model avoids overfitting, despite being trained on all noisy pixels.  This is a key advantage of zero-shot compression-based denoising compared to other zero-shot approaches.
>
> This claim is further verified in Figure 2, which shows  the PSNR of the denoised image over training iterations. It can be observed that the PSNR of ZS-NCD improves steadily and does not show degradation that is typical to overfitting. In contrast, in other zero-shot denoising methods overfitting is a known issue and is typically mitigated manually. For example,
>
> - DIP and Deep Decoder rely on manual early stopping or under-parameterization.
> - ZS-N2S, S2S and ZS-N2N modify the loss function to exclude some pixels, which is another form of manual regularization.
>
> In contrast, ZS-NCD does not require any of these manual heuristics and architectural constrains. In our approach, overfitting is prevented naturally by the compression-based  bottleneck that does not allow the model to reproduce the high-entropy noisy input image. We believe that this is a key advantage of our method.
>
> >Paper formatting concerns.
>
> We will update the manuscript to address all the formatting issues and have detailed explanations for the figures in the appendix.

---

> > ### Comment · Reviewer_t45e · 2025-08-04
> >
> > Thank you for addressing my questions with experimental results and detailed explanations.
> >
> > The author selectively chose some images (Kodak01, Kodak05)  to provide new experimental results, which do not comprehensively cover all images in Kodak24, so it does not fully convince me. Nevertheless, considering that this paper achieves state-of-the-art (SOTA) performance, I decide to raise my rating at the end.

---

> ### Author Response · Authors · 2025-08-07
>
> We thank the reviewer for recognizing the contribution of our work and deciding to raise the rating. Regarding generalizability, as we explained before, it is out of the scope of the zero-shot denoising problem. However, to address the reviewer’s comment, in our initial rebuttal we provided some simulation results that explored the generalization behavior of ZS-NCD by training the model on a noisy image and testing it on a different image.
>
> In the simulation results presented in our initial rebuttal (in response to Q3), we picked two random images, namely *kodim01* and *kodim05*, from the Kodak24 dataset. In the following tables, we provide a comprehensive comparison of the performance of ZS-NCD under two different settings.
>
> The first table reports the performance achieved by ZS-NCD when applied in the original zero-shot setting, i.e., trained and tested on the same image. In other words, for each image in the Kodak24 dataset, we first created its noisy version (according to the desired noise power) and trained a ZS-NCD model. Then, we denoised the same image using the trained model. The reported number is the average performance (PSNR and SSIM) achieved across all 24 images for $\sigma=15$ and $\sigma=50$.
>
> The following two tables present the average performance of ZS-NCD when there is a mismatch between the training and testing images. Column $i$ in the table, $i=1,\ldots,24$, corresponds to the average performance when a noisy version of image $i$ from the Kodak24 dataset is used for training. The reported average performance is computed on the remaining 23 images.
>
> It can be observed that, while the performance drops when ZS-NCD is used in this alternative setting,  typically the achieved performance remains reasonable.
>
> Our final table is a $24\times 24$ table. Let $i$ and $j$  denote the row  and  column indices, respectively. The $(i,j)$-th entry in this table is the PSNR achieved in denoising the $i$-th image in Kodak24 dataset when ZS-NCD model is trained on  image $j$ in the dataset. Here, the noise level is set as $\sigma = 15$. When comparing the entries within the $i$-th column, we observe that even when there is a mismatch between the training and test image  (i.e., $i \neq j$), the model still achieves reasonable denoising performance. Also, for each image $i$,  the highest PSNR in row $i$ corresponds to the model that is trained on image $i$, which aligns with the zero-shot setting.
>
> ||AWGN $(\sigma=15)$|AWGN $(\sigma=50)$|
> |:-:|:-:|:-:|
> |ZS-NCD (ours)|**33.18 / 0.9026**|**27.89 / 0.7464**|
>
> |$(\sigma=15)$ *kodim##*:|01|02|03|04|05|06|07|08|09|10|11|12|13|14|15|16|17|18|19|20|21|22|23|24|
> |:-:|:-:|:-:|:-:|:-:|:-:|:-:|:-:|:-:|:-:|:-:|:-:|:-:|:-:|:-:|:-:|:-:|:-:|:-:|:-:|:-:|:-:|:-:|:-:|:-:|
> |PSNR|27.54|25.75|30.38|27.95|31.51|27.23|29.52|26.95|30.04|26.79|28.99|28.98|26.94|31.32|30.30|26.01|27.70|31.18|28.46|27.29|28.21|29.98|30.41|29.62|
> |SSIM|0.8607|0.8505|0.8746|0.8723|0.8696|0.8655|0.8749|0.8540|0.8853|0.8607|0.8789|0.8647|0.8378|0.8720|0.8749|0.8633|0.8745|0.8730|0.8752|0.8713|0.8724|0.8797|0.8637|0.8731|
>
> |$(\sigma=50)$ *kodim##*:|01|02|03|04|05|06|07|08|09|10|11|12|13|14|15|16|17|18|19|20|21|22|23|24|
> |:-:|:-:|:-:|:-:|:-:|:-:|:-:|:-:|:-:|:-:|:-:|:-:|:-:|:-:|:-:|:-:|:-:|:-:|:-:|:-:|:-:|:-:|:-:|:-:|:-:|
> |PSNR|24.55|23.30|25.84|24.97|26.56|24.71|25.53|24.29|25.96|24.49|24.98|24.64|23.91|26.55|25.79|23.45|24.55|26.39|25.00|24.46|25.10|25.88|26.31|26.11|
> |SSIM|0.6838|0.7008|0.7080|0.6939|0.6731|0.6822|0.7002|0.6496|0.7141|0.7058|0.7079|0.6882|0.6262|0.6906|0.6925|0.6807|0.7085|0.6952|0.7138|0.7098|0.6983|0.7186|0.7020|0.7012|

---

> ### Author Response · Authors · 2025-08-07
>
> |$(\sigma=15)$|01|02|03|04|05|06|07|08|09|10|11|12|13|14|15|16|17|18|19|20|21|22|23|24|
> |:-:|:-:|:-:|:-:|:-:|:-:|:-:|:-:|:-:|:-:|:-:|:-:|:-:|:-:|:-:|:-:|:-:|:-:|:-:|:-:|:-:|:-:|:-:|:-:|:-:|
> |**01**|**31.28**|27.79|28.46|29.01|30.29|29.44|29.01|28.90|29.93|27.66|29.82|28.79|26.36|29.97|28.78|26.77|28.30|30.00|29.28|26.41|29.66|29.60|27.91|29.73|
> |**02**|29.05|**33.93**|32.38|28.20|32.06|30.66|32.99|24.60|29.02|22.58|32.54|30.38|15.86|32.95|31.93|18.00|21.87|31.02|30.12|19.08|24.78|29.92|31.90|24.75|
> |**03**|23.23|23.33|**35.87**|23.55|32.05|23.53|31.17|22.45|29.44|21.91|24.82|27.98|24.11|31.93|29.28|23.05|25.46|31.70|27.02|25.49|23.74|28.34|32.92|26.41|
> |**04**|26.64|26.28|32.26|**34.20**|32.67|26.15|31.25|26.63|30.34|28.08|28.44|31.08|19.92|32.10|32.44|21.23|23.50|30.86|27.31|21.12|25.89|31.95|32.56|27.68|
> |**05**|26.71|25.08|28.33|26.68|**31.64**|26.43|28.73|26.28|28.98|25.65|27.92|26.93|26.45|30.08|29.06|25.35|27.15|29.79|27.59|26.51|27.11|28.76|28.24|28.50|
> |**06**|29.01|28.13|29.97|26.64|30.93|**32.12**|29.81|27.33|30.33|27.67|29.97|29.21|28.56|30.90|29.30|29.08|28.04|30.54|28.15|27.86|29.49|30.11|28.75|30.41|
> |**07**|28.13|28.77|33.76|29.04|33.65|29.17|**35.36**|26.24|31.86|26.88|30.33|31.03|26.62|33.46|32.39|27.73|28.64|32.75|30.96|28.70|28.97|32.73|33.37|30.75|
> |**08**|26.69|23.06|24.51|26.30|28.44|24.85|25.37|**31.27**|27.71|27.04|27.26|26.18|27.04|27.64|27.26|24.65|26.76|28.32|25.82|26.67|27.02|27.74|26.24|28.28|
> |**09**|28.47|25.64|32.66|29.41|33.70|26.53|31.22|28.95|**35.06**|29.83|30.83|30.58|30.18|33.16|32.64|29.23|31.81|33.14|31.67|31.36|31.41|31.48|33.04|32.49|
> |**10**|30.49|26.62|31.62|32.59|33.55|27.63|31.26|31.40|33.41|**34.71**|32.35|30.84|31.07|33.05|32.95|30.85|31.86|33.15|31.20|31.78|32.52|31.82|32.64|33.10|
> |**11**|28.91|27.27|30.50|30.07|31.56|28.98|30.64|27.29|30.39|28.19|**32.40**|29.85|27.38|31.49|30.74|27.62|28.14|31.34|28.90|27.58|28.78|30.99|30.28|30.96|
> |**12**|30.91|26.76|33.05|30.68|33.19|29.71|30.60|29.85|30.97|30.50|30.04|**34.90**|29.52|32.81|33.43|28.79|29.53|32.21|27.73|27.76|30.08|30.59|32.59|32.23|
> |**13**|26.11|23.58|25.87|25.18|27.65|26.16|25.88|24.35|26.96|23.90|26.37|25.27|**28.61**|27.41|26.45|26.17|26.14|27.71|26.13|26.72|26.86|26.99|25.38|27.78|
> |**14**|24.26|23.68|30.08|23.26|31.13|24.64|29.00|23.33|29.33|21.92|25.06|27.22|25.17|**31.95**|28.49|22.91|26.59|30.08|28.33|26.66|24.84|27.75|29.39|26.36|
> |**15**|26.74|24.01|30.60|30.66|31.41|26.91|27.64|26.80|27.60|29.54|29.13|30.10|21.79|31.91|**34.39**|22.21|24.82|30.43|25.18|22.27|26.66|28.46|31.18|27.58|
> |**16**|31.58|28.70|32.88|31.09|32.84|29.41|31.91|30.69|33.14|31.05|32.72|31.66|32.19|32.88|31.58|**34.11**|32.04|32.67|32.43|32.30|33.04|32.26|31.41|32.82|
> |**17**|31.00|28.75|32.51|31.92|33.32|29.68|32.09|30.74|32.82|30.71|32.73|31.20|31.74|33.06|32.73|31.22|**34.08**|33.30|32.19|32.17|32.61|32.90|32.23|33.30|
> |**18**|26.89|26.15|28.70|26.50|30.34|27.63|29.29|25.22|29.46|24.67|28.17|27.75|27.33|29.92|29.36|26.65|27.91|**31.25**|28.95|27.93|27.90|30.04|28.70|29.90|
> |**19**|28.71|27.27|29.38|28.34|31.67|27.69|30.37|28.32|31.52|27.73|31.19|29.44|29.52|30.92|30.08|29.26|30.35|31.68|**33.27**|30.83|31.20|31.70|30.03|31.41|
> |**20**|25.51|24.98|29.77|27.42|29.81|29.35|26.93|27.78|29.60|27.28|27.17|30.39|31.24|32.26|31.36|25.11|29.75|31.91|24.47|**34.23**|27.85|29.73|32.05|30.47|
> |**21**|28.25|23.54|30.23|28.94|31.43|24.95|28.12|29.37|30.96|28.95|29.15|28.51|29.97|30.63|30.27|28.98|29.92|31.29|29.78|29.57|**32.45**|30.01|29.88|31.41|
> |**22**|26.49|25.63|30.32|27.56|31.49|26.62|30.18|24.95|30.74|26.39|29.10|29.17|27.15|30.67|30.57|26.36|27.37|31.59|28.78|27.66|27.85|**32.54**|30.76|30.65|
> |**23**|21.68|21.56|33.01|21.91|32.33|22.31|27.83|20.74|27.64|20.48|23.20|25.37|21.13|31.62|26.98|20.05|22.68|31.46|25.47|22.68|21.94|26.64|**36.02**|24.30|
> |**24**|27.99|25.64|27.78|27.93|29.33|27.83|27.60|27.63|28.72|27.46|28.47|27.57|29.27|29.47|28.78|27.02|28.54|30.23|27.18|28.66|28.56|29.14|27.98|**31.50**|

---

> > ### Comment · Reviewer_t45e · 2025-08-08
> > **Thanks for the response**
> >
> > Thank you for addressing my questions with experimental results. These responses have resolved most of my concerns. I will increase my original score.

---

### Decision · Program_Chairs · 2025-09-17

**Decision:**

Accept (spotlight)

**Comment:**

This paper proposes a new approach for "zero-shot" denoising (i.e., image denoising with machine learning models without requiring any prior training data) based neural compression. The authors provide a finite-sample theoretical guarantee in terms of an upper bound to the reconstruction error for their proposed approach for Gaussian and Poisson noise models. Experimental evaluation demonstrates the performance of their method, showing state of the art results.

**Strengths**
- Nice balance between theoretical analysis and strong-performing method.
- Intuitively simply approach based on information-theoretic notions.
- State of the art results for zero-shot denoising.

**Weaknesses**
- Somewhat limited and outdated baselines in their experiments.
- Some gaps between their theoretical analysis and the reported metrics in their results.
- Some parts of the presentation can be improved (e.g. in images in the appendix).

**Discussions and conclusion**

This paper received 3 reviews, and reviewers were mildly supportive of this paper initially. However, the discussion period was productive, and the authors have provided (i) further clarifications on their theoretical analysis, (ii) further experimental results on other datasets. This further increased the support. There's consensus that this paper represents an interesting and strong contribution to the field of image denoising in general, and it's worthy of publication - and I concur.